# Overexpression profiling reveals cellular requirements in the context of genetic backgrounds and environments

**Nozomu Saeki[1], Chie Yamamoto[1], Yuichi Eguchi[2], Takayuki Sekito[3], Shuji Shigenobu[4], Mami Yoshimura[5], Yoko Yashiroda[5], Charles Boone[5,6,7], Hisao Moriya[8]***

**1** Graduate School of Environmental and Life Science, Okayama University, Okayama, Japan, **2** Biomedical Business Center, RICOH Futures BU, Kanagawa, Japan, **3** Graduate School of Agriculture, Ehime University, Matsuyama, Japan, **4** National Institute for Basic Biology, Okazaki, Japan, **5** RIKEN Center for Sustainable Resource Science, Wako, Japan, **6** Donnelly Centre for Cellular and Biomolecular Research, University of Toronto, Toronto, Canada, **7** Department of Molecular Genetics, University of Toronto, Toronto, Canada, **8** Faculty of Environmental, Life, Natural Science and Technology, Okayama University, Okayama, Japan

* hisaom@okayama-u.ac.jp

**Data Availability Statement:** All relevant data are within the manuscript and its Supporting Information files.

## Abstract

Overexpression can help life adapt to stressful environments, making an examination of overexpressed genes valuable for understanding stress tolerance mechanisms. However, a systematic study of genes whose overexpression is functionally adaptive (GOFAs) under stress has yet to be conducted. We developed a new overexpression profiling method and systematically identified GOFAs in *Saccharomyces cerevisiae* under stress (heat, salt, and oxidative). Our results show that adaptive overexpression compensates for deficiencies and increases fitness under stress, like calcium under salt stress. We also investigated the impact of different genetic backgrounds on GOFAs, which varied among three *S. cerevisiae* strains reflecting differing calcium and potassium requirements for salt stress tolerance. Our study of a knockout collection also suggested that calcium prevents mitochondrial outbursts under salt stress. Mitochondria-enhancing GOFAs were only adaptive when adequate calcium was available and non-adaptive when calcium was deficient, supporting this idea. Our findings indicate that adaptive overexpression meets the cell's needs for maximizing the organism's adaptive capacity in the given environment and genetic context.

## Author summary

The study aimed to investigate how overexpression of genes can aid organisms in adapting to stress. The researchers utilized a new method to identify genes in yeast that demonstrated functional adaptability when overexpressed under stress such as heat, salt, and oxidative stress. The results indicated that overexpressing specific genes, like calcium-related genes, during salt stress could counteract deficiencies and improve the organism's ability to withstand stress. The study also examined the effect of different genetic backgrounds on these genes and discovered that the impact differed among various yeast strains.

**Funding:** This work was partly supported by Ohsumi Frontier Science Foundation, JSPS KAKENHI Grant Numbers 18H04824 20H03242, and 20H04870 (H.M.), 21J12451 (N.S.), and 17H06411 (C.B. and Y.Y). The funders had no role in study design, data collection and analysis, decision to publish, or preparation of the manuscript.

**Competing interests:** The authors have declared that no competing interests exist.

Additionally, the study revealed that calcium could play a key role in adapting to salt stress by preventing mitochondrial outbursts. These findings suggest that overexpressing certain genes can help the organism maximize its adaptability to stress in a given environment and genetic context.

## Introduction

Overexpression, namely excessive gene expression beyond the normal range of wild type, can be a beneficial resource for life to adapt to new environments. A well-established example is drug resistance. Cells and individuals acquire drug resistance when specific genes that interact with drugs are overexpressed [1]. For example, the glyphosate-resistant *Amaranthus palmeri* population from Georgia increased the 5-enolpyruvylshikimate-3-phosphate synthase gene to 5–160 copies [2]. Similarly, the malaria-causing parasite *Plasmodium falciparum* acquired resistance to the antimalarial drug Mefloquine through copy number variations (CNVs) of the *pfmdr1* gene [3]. Overexpression also confers adaptation to environmental stresses. Mutations that cause overexpression of HIF-1 (hypoxia-inducible factor) contribute to the adaptation of cancer cells to hypoxic environments [4,5]. An increase in the copy number of the *CUP1* gene, which encodes for the metallothionein protein, is associated with evolved copper tolerance in *Saccharomyces cerevisiae* [6]. Thus, overexpression is often observed as an adaptive phenomenon.

By artificially inducing overexpression, researchers can identify adaptive genes that can counteract growth inhibition caused by drugs, environmental stress, or mutations. *S. cerevisiae*, a commonly used model organism in eukaryotic research, is particularly useful in this approach due to its easily accessible overexpression plasmid libraries. The screening of drug targets using an overexpression library composed of random fragments from the *S. cerevisiae* genome was first carried out by Rine et al. in 1983 [7]. Subsequently, the technique has been refined and improved in later studies [8,9]. The overexpression screening also revealed genes that counteract environmental stress, such as *HAL1-9* under salt stress and *ZRC1* and *COT1* under heavy metal stress [10–17]. In addition, overexpression screening has been used to identify genes that counteract the adverse effects of deleterious mutations. The identification of such genes, known as "multicopy suppressors" (or dosage-dependent suppressors), was first performed by Bender and Pringle to isolate them for a cell polarity mutant *cdc24* [18]. Therefore, identifying genes whose <u>o</u>verexpression is <u>f</u>unctionally <u>a</u>daptive (GOFAs) and analyzing their functions are widely recognized as a powerful method that can provide insight into the mechanisms of drug resistance, stress resistance, and even normal cellular function [19].

Despite its benefits, the systematic exploration of GOFAs, especially in the context of wild types and under environmental stress, has never been performed. Traditional GOFAs screenings, as described above, were performed before the genomics era, and their comprehensiveness and reproducibility were not statistically evaluated. Notably, some studies, including ours, systematically identified genes whose overexpression is deleterious (i.e., dosage-sensitive genes) using systematic overexpression libraries where most of the annotated open reading frames (ORFs) of the *S. cerevisiae* genome are inserted and overexpressed [20–25]. However, those libraries have only been used in several cases to isolate beneficial genes [25]; namely GOFAs, and few studies have focused on beneficial genes. In addition, although the deleterious effect of overexpression in *S. cerevisiae* was tested among different strains [22], previous screenings for GOFAs primarily used a single "wild type" strain and did not consider variations among different strains. Recently, the diversity of *S. cerevisiae* strains has been recognized by

comparing thousands of isolates [26]. For example, the salt tolerance of strains varies greatly, and this difference can be explained by variations in the *PMR2* locus, where the sodium pump-encoding gene *ENA* is located [26,27]. A commonly used laboratory strain BY4741, derived from S288C, and the distantly related wine/European strain DBVPG6765 [28] have a cluster of three *ENA* genes. In contrast, the salt-sensitive laboratory strain CEN.PK [15] has only a single copy of the *ENA* gene (*ENA6*), which is weakly expressed. As suggested by multi-copy suppressor screening, adaptive (or suppressive) overexpression sometimes strongly associates with the genetic background; overexpression of a multicopy suppressor can have a positive function only in a mutant strain. Thus, the adaptability of overexpression can depend on genetic backgrounds (pre-existing variant sets), which is referred to as the context-dependency [29,30]. Therefore, the differences in genetic background between strains, as described above, could have an impact on identifying GOFAs under salt stress; however, this has not been considered in previous studies.

In this study, we aimed to systematically identify GOFAs using a novel overexpression profiling approach in *S. cerevisiae*. By doing so, we aimed to provide a comprehensive and reproducible method for identifying GOFAs and to gain deeper insight into the mechanisms of stress resistance via overexpression. Furthermore, by considering the genetic background of different strains, we tried to uncover context-dependent GOFAs that may have varying effects on stress resistance in different strains. In overexpression profiling, we competitively cultivated pooled overexpression libraries that covered over 93% of the genome and evaluated the occupancy using high-throughput sequencing. Isolated GOFAs under three well-studied stress conditions: high temperature, high salt, and oxidative stress, suggested that adaptive overexpression can compensate for deficiencies to maximize fitness under stress. The difference in GOFAs identified in BY4741, CEN.PK and DBVPG6765 suggested that the adaptiveness of overexpression is highly dependent on both the genetic background and environmental conditions. Moreover, we showed that the overexpression profile could determine the missing piece for cell growth in each genetic background and environment.

## Results

### GOFAs under stress are a distinct set of genes from stress-resistant genes

To systematically identify GOFAs, we set up an experimental system called "overexpression profiling," as shown in **Fig 1A**. The system consists of four steps: 1) construction of pools of *S. cerevisiae* cells harboring a multicopy 2μ plasmid containing each gene in the genome; 2) competitive culture and passaging of the pooled cells; 3) long-read sequencing of plasmid inserts extracted from the cultured cells; and 4) analysis of sequence data to calculate the occupancy of plasmid inserts to identify GOFAs. In the initial version of this experiment, we pooled BY4741 cells harboring a 2μ plasmid with each of 5,751 genes in the BY4741 genome. The two independent sets of the overexpression strains were previously prepared [21] and maintained as a frozen stock in a 96-well format. We pooled each independent stock to construct Pool_a and Pool_b, respectively. An example of mapped insert reads is shown in **S1A Fig**. The initial Pool_a and Pool_b both covered over 93% of the 5,751 genes (**S1B Fig**), suggesting that this experiment should be highly comprehensive.

As a proof of concept for the overexpression profiling, we first tried identifying GOFAs in the presence of 250 μM methotrexate (MTX), an antagonist of dihydrofolate reductase encoded by *DFR1* in yeast [31]. As expected, the *DFR1* reads were enriched to over 90% of the total reads after 30 generations of competitive culture with MTX in both Pool_a and Pool_b (**Fig 1B**), supporting that our system can effectively identify GOFAs. We next attempted to identify GOFAs under heat, salt/osmotic, and oxidative stresses as they are among the most

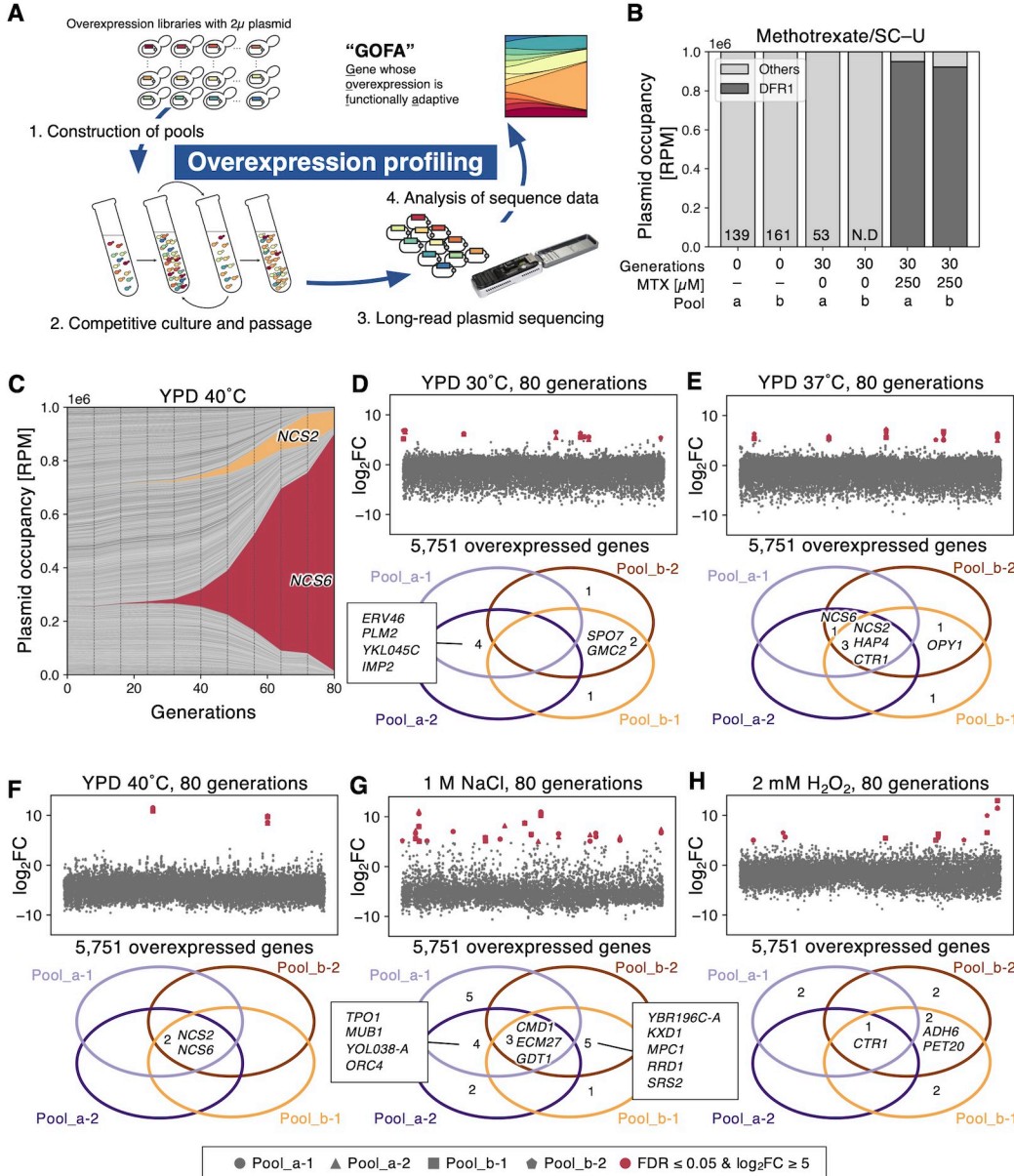

**Fig 1. Isolation of GOFAs by overexpression profiling. A)** "Overexpression profiling" for identifying GOFAs developed in this study. The detail is explained in the text. **B)** A proof of concept for overexpression profiling: identification of GOFA under 250 μM methotrexate. The bar plot and the numbers on the bars show occupancies of the *DFR1* with reads per million reads (RPM). **C)** The time course of plasmid occupancy under heat stress. One of the four replicates (Pool_a-2) at 40˚C in YPD for 80 generations (samples were analyzed every eight generations). Occupancies of each plasmid are shown with reads per million reads (RPM). The orange and red areas correspond to *NCS2* and *NCS6* reads, respectively. **D-H)** Fold changes of plasmid occupancies after the cultivation (upper, FDR ≤ 0.05 and $\log_2$FC ≥ 5) and Venn diagrams of hits in replicates (lower, FDR ≤ 0.05 and $\log_2$FC ≥ 5) under well-studied stresses; YPD at 30˚C (**D**, control), 37˚C (**E**) and 40˚C (**F**) under the heat stresses, 1 M NaCl (**G**) as the salt stress, 2 mM $H_2O_2$ (**H**) as the oxidative stress. The $\log_2$FC is plotted along the y-axis as a function of the 5,751 overexpressed genes ordered by ORF names. Hits were shown as red-filled symbols. Hit genes are summarized in **S2 Table**.

studied stresses in yeast (**S2A Fig**). We selected 37˚C/40˚C, 1 M NaCl, and 2 mM $H_2O_2$ as stress conditions after conducting preliminary growth experiments (**S2B–S2D Fig**).

To evaluate the reproducibility of the experiments, we performed four replications (i.e., two using each of Pool_a and Pool_b). We believe that repeating experiments is crucial in

distinguishing between adaptive mutant cells that emerged randomly during competitive culture and adaptive cells that overexpress specific genes that were present in the pool from the start. Specifically, if the same overexpression strain is enriched after multiple independent cultures, it strongly suggests that the overexpression strain was adaptive rather than resulting from mutation. Replications should also be important to identify primary mutations affecting a gene unrelated to the overexpressed one. For the heat stress, we also obtained the time course of plasmid occupancy (**Figs 1C** and **S3**). Overall, plasmid reads from replicates gave satisfactory reproducibility (Pearson's $r > 0.570$), but there was a higher correlation between the duplicates (**S3E and S3G Fig**) from the same pool than between duplicates from different pools (**S3F and S3H Fig**). This suggests that some biases may be present in the isolated GOFAs depending on the differences between the two pools. Additionally, there was a negative correlation between the insert lengths and the number of reads (**S4A–S4E Fig**), indicating a technical bias in the experiment that selectively detects shorter inserts. To reduce these biases, we calculated the fold change (FC) in the plasmid occupancy of each gene over the occupancy in the initial pool, and then defined genes with the $\log_2\text{FC} \geq 5$ and the false discovery rate (FDR) $\leq 0.05$ as "hits" (see Materials and Methods). **Fig 1D–1H** show the fold change of 5,751 genes and the hits in each condition. There were still higher correlations in the fold change between replicates from the same pools than different pools (**S4F–S4M Fig**). Therefore, the fold-change normalization did not eliminate biases created by the difference between two pools.

After 80 generations in control YPD at 30°C conditions, more than 4,500 out of 5,751 genes (78%) were still detected (**S3F Fig**), suggesting that the plasmids were quite stable even in the absence of plasmid maintenance selection. As expected, the dosage-sensitive genes identified in our previous studies, those for which multiple copies have a negative impact on proliferation [21], were dropped out of the culture (**S5A–S5C Fig**). No hits were shared among the four replicates (**Fig 1D**), indicating that there is no dominant GOFA in this condition. In the stress conditions, we identified *NCS2*, *CTR1*, and *HAP4* at 37°C; *NCS2* and *NCS6* at 40°C; *CMD1*, *ECM27* and *GDT1* under 1 M NaCl; and *CTR1* under 2 mM $H_2O_2$, as hits in all four replicates (**Fig 1E-1H**). The permutation test demonstrated that the observed overlaps of genes being hit in 2/4, 3/4, and 4/4 replicates were statistically significant ($p < 0.05$, S9 Table), indicating a non-random pattern in the distribution of gene hits across the replicates. However, the overlap of genes being hit in 1/4 replicates was not found to be significant ($p = 1.00$, S9 Table). These findings support the claim that the overlapped genes are likely to be GOFAs.

We initially expected that our overexpression profiling would identify so-called "stress-responsive genes" induced under stress conditions. For example, we thought that overexpression of heat shock-induced chaperones might confer high-temperature tolerance. However, none of the GOFAs matched stress-inducible genes such as "environmental stress-responsive genes" or "heat shock-responsive genes" [32,33]. Although the original libraries contained these stress-responsive genes (**S5D Fig**), they were eliminated during the competitive culture (**S5E–S5H Fig**). These results suggest that the identified GOFAs are a distinct set of genes not induced under stress but whose overexpression is adaptive.

To gain insights into the nature of the identified GOFAs, we first focused on *NCS2* and *NCS6*, which were identified as confident GOFAs at 40°C (**Fig 1C and 1F**). To verify that they are GOFAs under heat stress, we independently cultivated BY4741 overexpressing (*-oe*) those genes. It is worth noting that, to verify the function of GOFAs, we employed the use of freshly prepared plasmid transformants that had not been exposed to stress conditions in all experiments. This approach was taken to confirm that any observed phenotypes were a direct result of the overexpression of specific genes rather than mutations that may have occurred during adaptation to stress or storage. *NCS6-oe* significantly increased the growth rate by 1.29-fold at 40°C. *NCS2-oe* increased the growth rate by 1.14-fold at 40°C, although the increase was not

statistically significant (**S6 Fig**). *NCS2* and *NCS6* are involved in the thiolation of wobble uridine of tRNAs in the *URM1* pathway [34], and the *URM1* pathway is known to have strain-dependent thermosensitivity [35,36]. Since the BY4741 strain derived from S288C has a thermosensitive *URM1* pathway [36], we hypothesized that *NCS2-oe* and *NCS6-oe* would compensate for their functions at high temperatures. Therefore, we posited that GOFAs generally function to compensate for cellular requirements in each environment.

## GOFAs enriched under salt stress propose $Ca^{2+}$ limitation of the culture medium

To further confirm our hypothesis, we next focused on GOFAs enriched under salt stress (1 M NaCl, **Fig 1G**). *CMD1*, *GDT1*, and *ECM27* were enriched in all four replicates (**Fig 2A**). *CMD1* encodes calmodulin [37], and *GDT1* and *ECM27* encode calcium transporters that localize to the Golgi and ER membranes, respectively [38,39]. They are thus all involved in intracellular calcium homeostasis (**Fig 2B**). We confirmed that their overexpression significantly increased the growth rate under 1 M NaCl (**Fig 2C**). In addition to these three GOFAs, we also analyzed *YBR196C-A*, which was a hit only in the Pool_b-derived replicates (**Fig 2A**). *YBR196C-A* is recognized as an "emerging gene" and encodes an adaptive protein that localizes to the ER membrane (**Fig 2B**) [40], while its concrete function has not been revealed yet. We also confirmed that *YBR196C-A-oe* significantly increased the growth rate under 1 M NaCl (**Fig 2C**), indicating that genes only hit in each replicate might also contain GOFAs.

We confirmed that overexpression of *GDT1* and *ECM27* enhanced the "calcium pulse" (rapid increases in cytosolic $Ca^{2+}$ concentration) upon salt/osmotic stress exposure [38,39,41,42] (**Fig 2D**). *YBR196C-A-oe* also enhanced the calcium pulse even more strongly (**Fig 2D**). Based on these results, we speculated that GOFAs under high salt might compensate for the calcium requirement under our experimental conditions. This was indeed the case. The addition of $Ca^{2+}$ up to 20 mM in a 1M NaCl medium was found to enhance the growth rate of cells under salt stress, as shown in **Fig 2E**. However, further increases above 50 mM led to a decrease in growth. Furthermore, the increase in growth rate with 5 mM $Ca^{2+}$ addition negated the advantage of GOFAs under salt stress (**Fig 2F**), suggesting that GOFAs mimic $Ca^{2+}$ addition.

During the competitive culture under salt stress, we unexpectedly observed that the control strain without libraries also adapted (or "evolved") to the salt stress (lineage1 and lineage2 in **Fig 2G**). The two evolved lineages after 10 passages (Ev) grew significantly faster than the ancestral strain (An) under 1 M NaCl (**Fig 2H and 2I**). We performed whole-genome sequencing on pooled samples of the An and Ev strains and identified a total of six mutations (two on lineage 1 and four on lineage 2, **S3 Table**). Four of the six mutations were identified in the coding regions of *PMA1*, *PMR1*, and *TRZ1*. One of these mutations, located in *TRZ1*, was also present in the ancestor. The other two mutations were found in intergenic DNA. A mutation in lineage 1 caused a P393Q substitution in Pma1, and two mutations in lineage 2 resulted in G428D and G824E changes in Pmr1 (**Fig 2J**). Since Pma1 and Pmr1 are also involved in calcium homeostasis [43,44] (**Fig 2B**), we speculated that cells may have also evolved to compensate for calcium deficiency under our conditions through these mutations. This idea also seemed correct since the increased growth rate due to the addition of $Ca^{2+}$ under salt stress negated the advantage of the Ev strain (**Fig 2H and 2I**, right).

The results so far indicate that the growth conditions used here lack sufficient calcium for maximum salt tolerance of the yeast cells, resulting in the isolation of genes that may compensate for the calcium requirement by overexpression (or mutation).

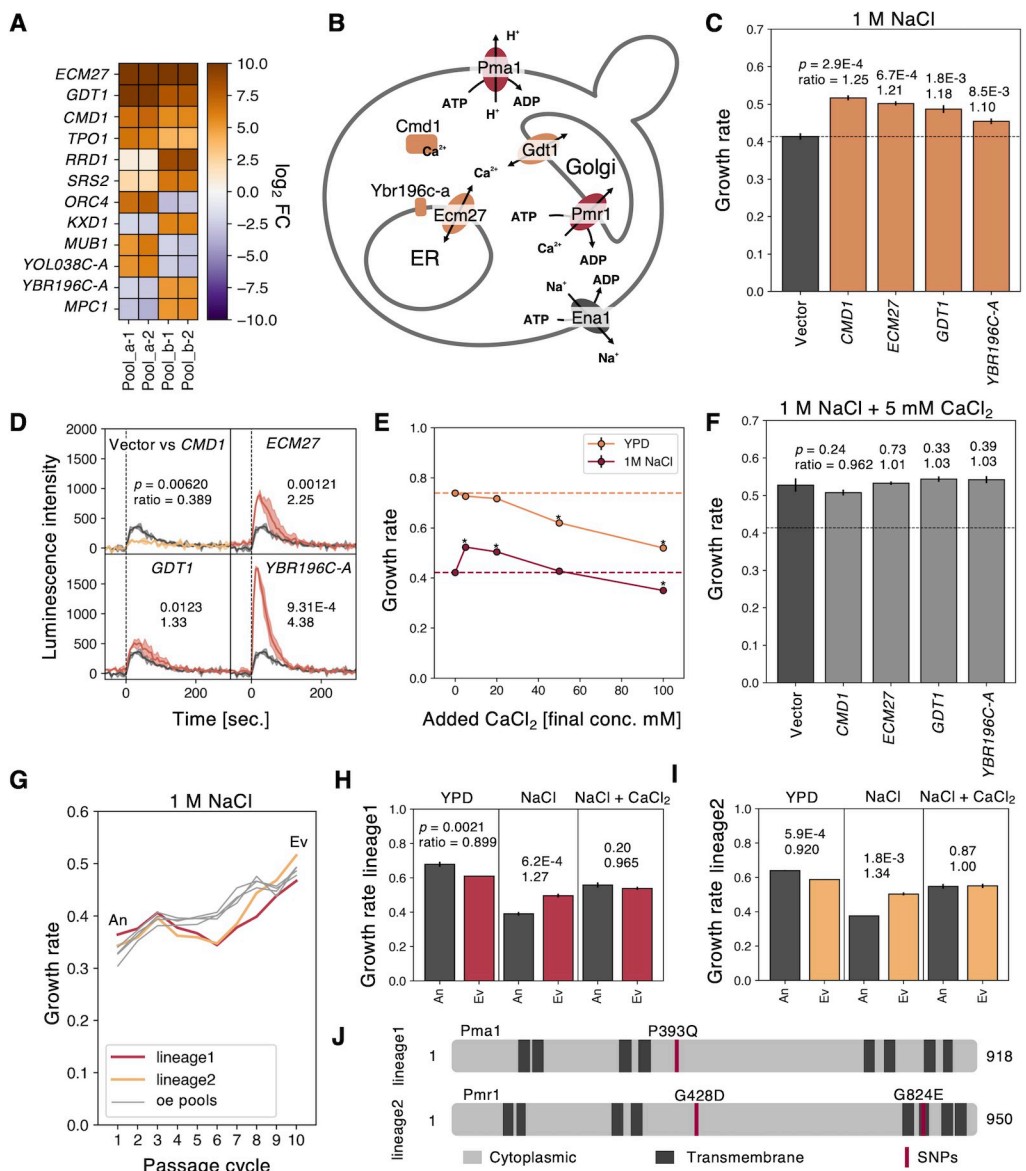

**Fig 2. GOFAs as well as adaptive mutants under salt stress propose Ca²⁺ limitation in the culture medium. A)** Genes whose $\log_2$FC was $\geq 5$ and FDR $\leq 0.05$in at least two of four replicates under salt stress. Their $\log_2$FC values are shown with color. **B)** A cellular diagram illustrating protein functions in calcium homeostasis identified in this study. **C)** Growth rates of the cells overexpressing GOFAs under 1 M NaCl. The dashed line presents growth rate of the empty vector control. **D)** The cytoplasmic Ca²⁺ pulses of the cells overexpressing GOFAs upon salt stress measured by the aequorin luminescence assay. The dashed lines represent the timing of NaCl addition. Each maximum value of luminescence intensity was used for Welch's t-test. **E)** The effect of CaCl₂ addition on growth rates of BY4741 with or without 1 M NaCl. The dashed lines show the growth rate of BY4741 in YPD or 1 M NaCl without CaCl₂ addition. Asterisks indicate significant differences. **F)** Growth rates of the cells overexpressing GOFAs under 1 M NaCl with 5 mM CaCl₂. The dashed line presents growth rate of the empty vector control without CaCl₂ addition in **C**. **G)** Growth rates of overexpression (oe) pools and the empty vector controls during the passages under salt stress. The linage1 and linege2 of the vector controls and four oe pools under 1 M NaCl are shown. For the vector control, the cells first inoculated were designated as "ancestor (An)" and the cells obtained after the 10th passage cycle as "evolved (Ev)". **H** and **I)** Growth rates of An and Ev cells of linege1 (**H**) and linege2 (**I**) under YPD, 1 M NaCl, and 1 M NaCl with 5 mM CaCl₂. **J)** Diagrams showing the amino acid substitutions in Pma1 and Pmr1 from the Ev lineage1 and lineage2. The dark gray areas indicate the transmembrane domains, and the red bars indicate amino acid substitutions. The *p*-values are from Welch's t-test ($n = 3$). The significance was evaluated by the Bonferroni correction ($p \leq 0.05/4 = 0.0125$). Error bars or the filled areas indicate SD.

## GOFAs reflect differences in yeast strains

Previous studies have found that *ENA1* and *HAL1-9* confer salt tolerance through copy number variation or overexpression [10–15,27]. We were thus surprised that the GOFAs under the salt stress did not include those genes (**Figs 1G** and **2A**), even though they were present in our original libraries (**S1 Data**). Because *S. cerevisiae* varies greatly in salt tolerance among strains [26] (**S7A Fig**), we speculated that differences in the genetic background might affect the identified GOFAs.

To test this possibility, we analyzed the effect of calcium on the salt tolerance of different *S. cerevisiae* strains, namely BY4741, W303, CEN.PK2-1C (CEN.PK), and DBVP6765 (**Fig 3A**). We also analyzed several other strains (**S7B Fig**). As reported, their salt tolerance without calcium was quite different; under 1 M NaCl, DBVP6765 grew much slower than BY4741 and W303, and CEN.PK did not grow (**Fig 3A**, 0 mM Ca$^{2+}$). Interestingly, adding calcium dramatically improved the salt tolerance of DBVP6765 and CEN.PK; adding 5 mM calcium nearly canceled the salt sensitivity of DBVP6765 compared to BY4741 and W303 (**Fig 3A**), while the salt tolerance of CEN.PK increased gradually but significantly with the addition of calcium up to 50 mM (**Fig 3A**). Note that the addition of calcium did not increase the growth rate of the strains without salt stress (**S7C and S7D Fig**). These results suggest that the differences in salt sensitivity of each strain are explained by differences in calcium requirements, which could potentially reflect differences in GOFAs. Therefore, we next attempted to identify the GOFAs of CEN.PK and DBVP6765 under salt stress.

We developed a method to transfer the plasmid library to other strains. We performed homologous recombination by using a mixed PCR product containing the 5,803 genes of the BY4741 genome and 2μ plasmids in yeast cells (**Fig 3B**). The constructed pooled libraries of CEN.PK (Pool_C) and DBVPG6765 (Pool_D) covered more than 5,000 genes (**Fig 3C**).

We then performed the overexpression profiling using three replicates from each of Pool_C and Pool_D. Both the Pool_C and Pool_D strains grew faster and adapted more rapidly to salt stress compared to the vector control (**S8A Fig**). We shortened the duration of the competitive culture for CEN.PK and DBVPG6765 in comparison to BY4741, terminating it after 8 and 16 generations, respectively. This is because we consider that adaptive plasmids should already be enriched in these generations. As expected, the diversity of the pooled library, as measured by the Gini-Simpson index, decreased quickly when using CEN.PK and DBVPG6765, as opposed to BY4741 (**Fig 3D**).

The genes that were identified as hits in each strain are presented in **Fig 3E-3G**. Seventeen genes in CEN.PK and 13 genes in DBVPG6765 were hit in all the triplicate experiments. The permutation test demonstrated that the observed overlaps of genes being hit in 2/3 and 3/3 replicates were statistically significant (CEN.PK hit in 2/3 $p < 0.0001$, hit in 3/3 $p < 0.0001$ and DBVPG6765 hit in 2/3 $p < 0.0001$, hit in 3/3 $p < 0.0001$), indicating a non-random pattern in the distribution of gene hits across the replicates. However, the overlap of genes being hit in 1/3 replicates was not found to be significant (CEN.PK hit in 1/3 $p = 1.00$ and DBVPG6765 hit in 1/3 p = 1.00). These findings support the claim that the overlapped genes are likely to be GOFAs. To compare potential GOFAs among the strains, we utilized genes with significant hits using the permutation test (CEN.PK and DBVPG6765 $\geq$ 2/3, BY4741 $\geq$ 2/4). Additionally, to maintain consistency in diversity, we used hits identified in 48 generations of BY4741 (as represented by the large circles in **Fig 3D**). Interestingly, most of the identified genes were specific to a single strain, and only a few genes appeared in more than two strains. *SAT4/HAL4* was an overlapping hit between BY4741 and CEN.PK (**Figs 3H** and **S8**). *ECM27*, *GDT1*, and *CMD1* overlapped between BY4741 and DBVPG6765. *SIS2/HAL3*, *HAL5*, and *CRZ1/HAL8* overlapped between CEN.PK and DBVPG6765. Notably, *ENA1* was only identified in CEN.

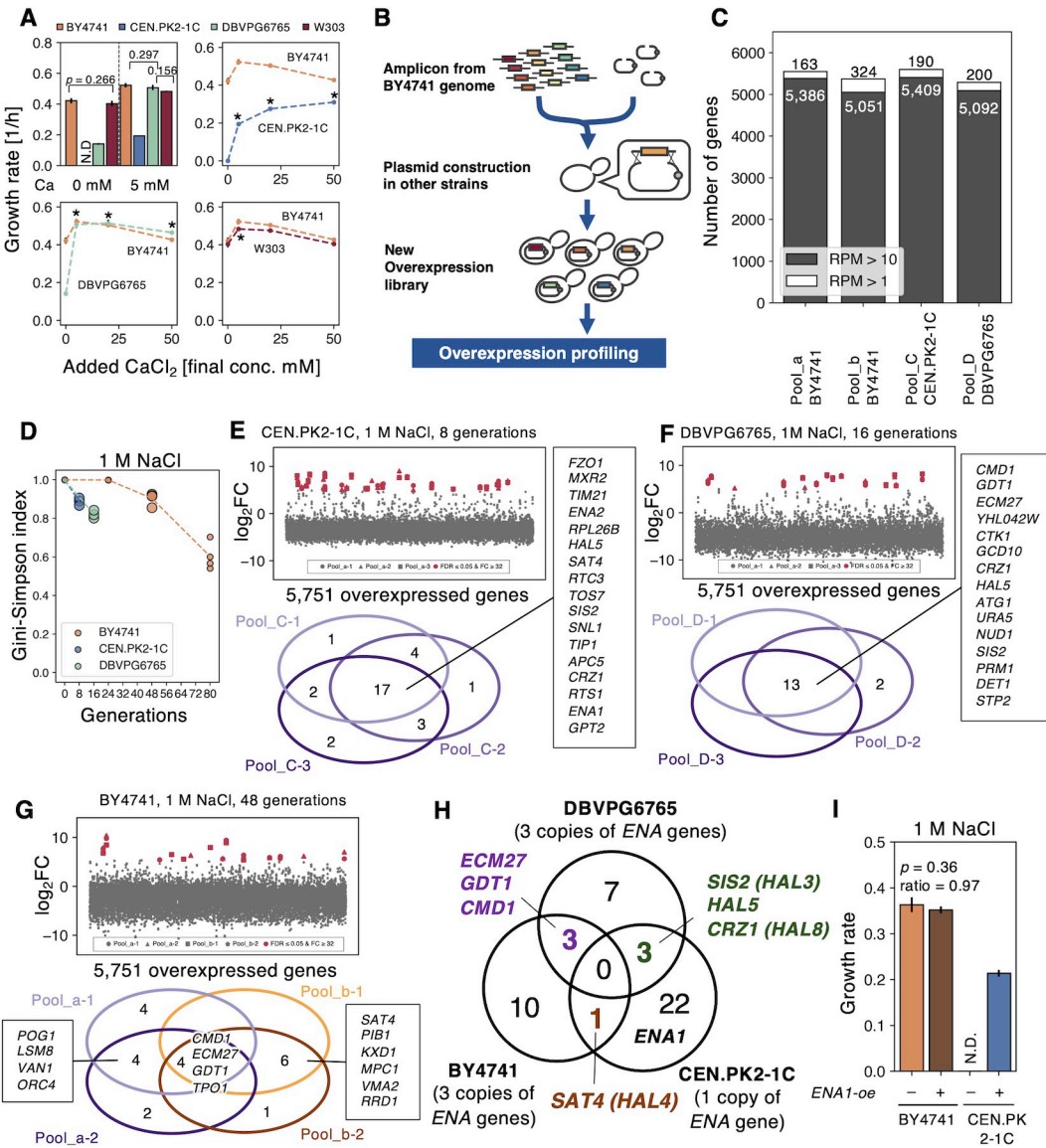

**Fig 3. GOFAs reflect differences in genetic backgrounds. A)** Relationship between CaCl$_2$ addition and growth rates of various *S. cerevisiae* strains. Growth rates indicate strains under 1M NaCl with added CaCl$_2$ are shown. The growth rate of CEN.PK2-1C without CaCl$_2$ addition (no growth) was set to 0 for convenience. The pairs without *p-value* indicate FDR $\leq$ 0,05 using Welch's t-test and Benjamini-Hochberg method in the upper left panel. Asterisks mean significant difference against those without Ca$^{2+}$ addition (p$\leq$0.05/4). **B)** The new overexpression libraries constructed in this study. The detail is explained in the text. **C)** Coverages of the constructed libraries. The filled bars indicate RPM $\geq$ 10, and the unfilled bars indicate RPM $\geq$ 1. **D)** Decrease in the diversities of plasmids in the pooled libraries of CEN.PK2-1C, DBVPG6756, and BY4741 during the cultivation under salt stress. The diversity was evaluated as the Gini-Simpson index. The large circles indicate the data points used in **H**. **E-G)** Fold changes of plasmid occupancies after the cultivation (upper) and hit genes in each replicate 1 M NaCl (lower, FDR $\leq$ 0.05 and log$_2$FC $\geq$ 5). Hit genes were shown as red-filled symbols. CEN.PK2-1C with 8 generations (**E**), DBVPG6765 with 16 generations (**F**), and BY4741 with 48 generations (**G**). The overexpression profiling of CEN.PK2-1C and DBVPG6765 were performed with three replicates originating from Pool_C and Pool_D, respectively, and BY4741 with four replicates from Pool_a and Pool_b. The log$_2$FC is plotted along the y-axis as a function of the 5,751 overexpressed genes ordered by ORF names. Hits are summarized in [S4 Table](). **H)** A Venn diagram showing overlaps of hit genes in BY4741, DBVPG6765, and CEN.PK2-1C. **I)** Growth rates of *ENA1-oe* in BY4741 and CEN.PK2-1C under salt stress. The *p*-values are from Welch's t-test (*n* = 3). Error bars represent SD (*n* = 3).

PK. Indeed, *ENA1-oe* in CEN.PK restored growth under 1 M NaCl but not in BY4741 (**Fig 3I**). These results suggest that the differences in genetic background may explain why genes that had been identified in previous studies were not identified in the initial overexpression profiling on BY4741. In addition, this overexpression profiling has the potential to reveal genetic background-dependent requirements.

## GOFA reflects the factors that the strain requires in each environment

The difference in the *ENA* gene copy [15,28,45,46] (**S9 Fig**) may explain why *ENA1* was only isolated as a hit in CEN.PK, and why there are differences in GOFAs under salt stress (**Fig 3H**). Therefore, we investigated how GOFAs are affected when sufficient *ENA* function is provided to CEN.PK through *ENA1*-overexpression. To accomplish this, we created a pooled library of diploid CEN.PK2 (Pool_CE) by crossing the pooled library constructed in CEN. PK2-1C (Pool_C) with an *ENA1*-overexpressing CEN.PK2-1D (referred to as *ENA1-coe*, **Figs 4A** and **S10**).

**Fig 4B** shows the fold change of genes in the *ENA1-coe* pool after 16 generations under salt stress. Thirteen genes were hit in all the triplicate experiments. The permutation test demonstrated that the observed overlaps of genes being hit in 2/3 and 3/3 replicates were statistically significant (hit in 2/3 $p < 0.0001$, and hit in 3/3 $p < 0.0001$), indicating a non-random pattern in the distribution of gene hits across the replicates. As expected, *ENA1*-overexpression altered the hits (**Fig 4C**). *ENA1* itself, *CRZ1*, and *SIS2* were no longer in GOFAs, suggesting that their functions are directly related to the *ENA* function. The gene for calcium homeostasis, *ECM27*, became a GOFA, indicating that calcium may be a secondary requirement for optimal *ENA* function. The most enriched GOFAs under both with and without *ENA1*-overexpression were *SAT4* and *HAL5*, which encode protein kinases that regulate potassium importers [13]. Since these genes were isolated independently from enhanced *ENA* functions (**Fig 4C**), we inferred that there might be another requirement for CEN.PK other than *ENA1*, which is potassium.

Indeed, as previously reported [47], the addition of potassium ($K^+$) increased the salt tolerance of CEN.PK (**Fig 4D**). The addition of $K^+$ also increased the growth rate under *ENA1*-overexpression, and the addition of both $K^+$ and calcium ($Ca^{2+}$) further increased the growth rate (**Fig 4D**), confirming that potassium is required for the salt tolerance of CEN.PK, in addition to calcium and enhanced *ENA* function. If GOFAs reflect the factors a strain requires in different environments, the differences in salt-tolerant GOFAs between BY4741 and CEN. PK (**Fig 3H**) should indicate variations in the amount of calcium and potassium required for salt tolerance. To test this, we measured the growth rates of BY4741 and CEN.PK under 1 M NaCl conditions at varying concentrations of $Ca^{2+}$ and $K^+$. As expected, the fitness landscapes of the two strains were distinct (**Fig 4E and 4F**): BY4741 grew best at 5 mM $Ca^{2+}$ and 100 mM $K^+$, while CEN.PK grew best at 50 mM $Ca^{2+}$ and 500 mM $K^+$. We believe that these requirements are mimicked by overexpressing specific genes, such as *ECM27*, *GDT1*, and *CMD1* in BY4741 (**Fig 4G**) and *ENA1*, *ECM27*, *SAT4*, and *HAL5* in CEN.PK (**Fig 4H**). We confirmed the $Ca^{2+}$ requirement of CEN.PK is mimicked by overexpressing *ENA1* and *ECM27*, which additively confer salt tolerance, but the effect is less pronounced when $Ca^{2+}$ is added (**Fig 4I and 4J**).

It's worth noting that the $Ca^{2+}$ and $K^+$ requirements for salt tolerance may reflect the natural conditions of the yeast. Specifically, the salts found in nature contain potassium and calcium, and when these natural salts were used, yeast growth was better than when pure NaCl was used (**S11 Fig**). In other words, the growth landscapes shown in **Fig 4E** and **4F** are likely optimized for the specific salt composition of the living environment of each strain.

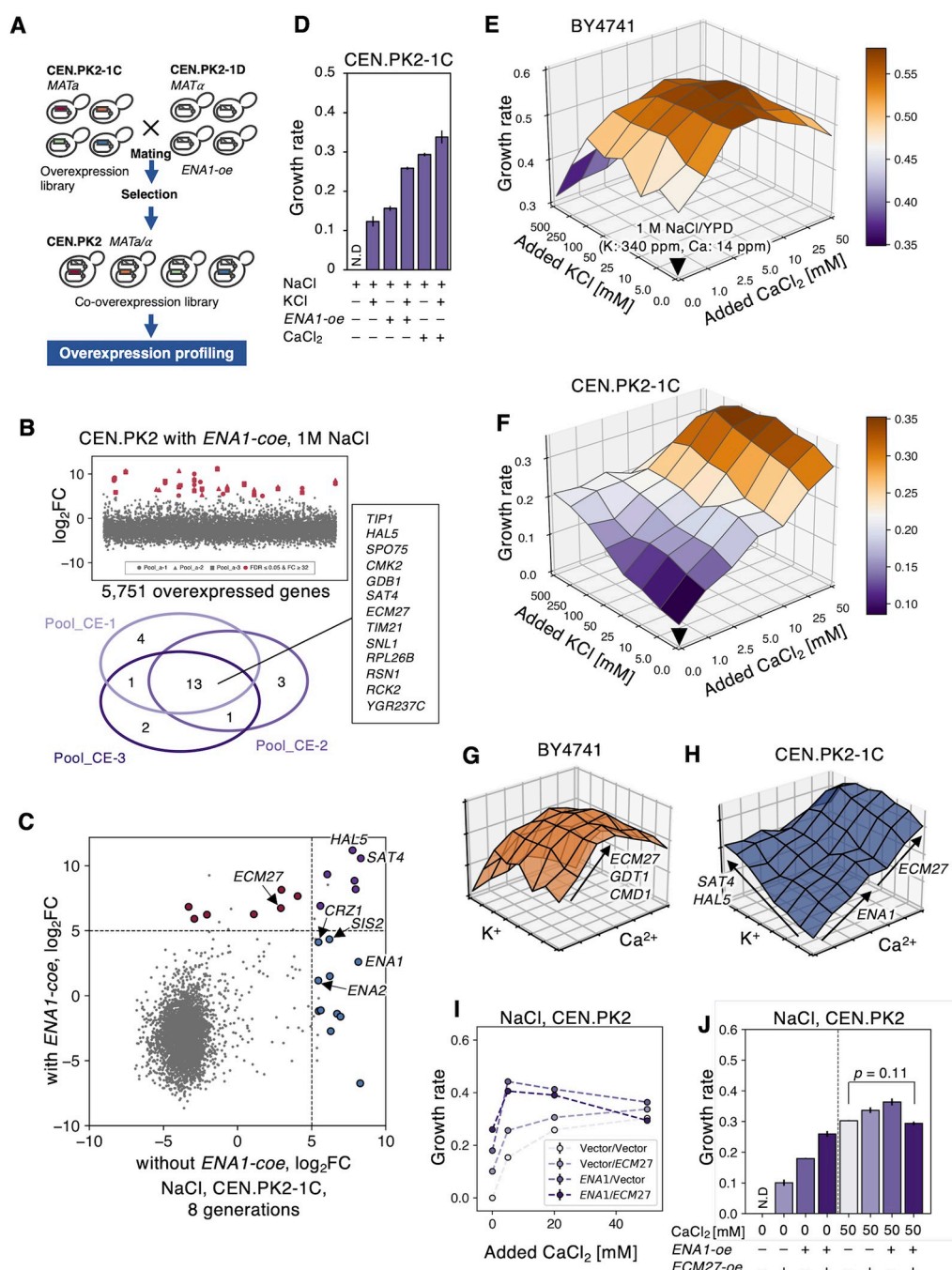

**Fig 4. Strain-dependent requirements of calcium and potassium for the salt stress reflect strain-dependent GOFAs. A)** Construction of the *ENA1* co-overexpression (*-coe*) library by mating. The detail is explained in the text. **B)** Fold change of plasmid occupancy after the 16 generations in CEN.PK2 with *ENA1-oe* under 1 M NaCl (upper), and a Venn diagram of hits in replicates (lower, FDR ≤ 0.05 and $\log_2$FC ≥ 5). The $\log_2$FC is plotted along the y-axis as a function of the 5,751 overexpressed genes ordered by ORF names. These data are summarized in **S4 Table**. **C)** A comparison of the mean fold change of plasmid occupancy with and without *ENA1-coe* under 1 M NaCl. The colored circles indicate triple hit genes in three replicates: without *ENA1-core* (blue), with *ENA1-core* (red), and both (purple). The dashed lines represent the threshold of hits as $\log_2$FC ≥ 5. **D)** Growth rates of CEN.PK2-1C under 1 M NaCl with supplements. N.D means not detected. Error bars indicate SD. All 15 pairs differed significantly (Welch's t-test and Benjamini-Hochberg correction, FDR ≤ 0.05, *n* = 3). The value of N.D is set to 0 for the statistical test. **E-F)** Fitness landscapes of BY4741 (**E**) and CEN.PK2-1C (**F**) under 1 M NaCl with various KCl and CaCl₂ levels. The downward triangle points to 1 M NaCl/YPD, with increasing amounts of KCl or CaCl₂, added along the *x*- or *y*-axes. The growth rates at each KCl and CaCl₂ addition are represented as the *z*-axis and colored as a purple-to-orange heat map,

corresponding to the relative growth rate. **G-H)** A diagram of the expected relationship between slopes on fitness landscapes and GOFAs in BY4741 (**G**) and CEN.PK2-1C (**H**). Arrows indicate the correspondence between $Ca^{2+}$ or $K^+$ requirement and each GOFA. **I and J)** Effects of $CaCl_2$ addition on the growth rates of CEN.PK cells overexpressing *ENA1* (*ENA1-oe*) and *ECM27* (*ECM27-oe*). *ENA1* and *ECM27* were overexpressed using pTOW48036 and pRS423nz, respectively. The Vector/Vector cells without $CaCl_2$ addition did not grow, but the growth rate was set to 0 for convenience in **I** and shown as N.D in **J**. Error bars indicate SD ($n = 3$). All 6 pairs with 0 mM $CaCl_2$ and 5 pairs with 50 mM $CaCl_2$ were significantly different (Welch's t-test and Benjamini-Hochberg correction, FDR $\leq$ 0.05, $n = 3$). A pair with no significance is shown in the figure. The value of N.D is set to 0 for the statistical test.

## Mitochondria appear to be the primary target for enhanced salt tolerance with calcium addition

We next attempted to uncover how calcium confers salt tolerance. It is believed that salt tolerance and calcium in *S. cerevisiae* are connected to a short-term stress response, in which a transient $Ca^{2+}$ pulse triggered by the salt stress leads to the induction of *ENA1* [48] (as shown in **S12A Fig**). However, $Ca^{2+}$ addition did not increase Ena1 induction under 1 M NaCl in BY4741(**Fig 5A**). Even in deletion mutants of components of the calcium-dependent *ENA1* induction pathway (*CNB1* and *CRZ1*), $Ca^{2+}$ addition increased salt tolerance (**S12B Fig**). Calcium seems to act as a sustainable effector rather than a transient signaling activator, as adding calcium restored growth retardation long after salt stress exposure, while adding calcium prior to stress did not improve growth (as shown in **S12C–S12F Fig**). Therefore, there must be another mechanism for calcium-dependent salt tolerance other than the short-term stress response.

To understand the unknown mechanism by which calcium confers salt tolerance, we used a complementary approach to overexpression profiling: functional profiling of gene knockout mutants. We competitively cultured a pooled knockout collection [49] and performed relative fitness analysis under three conditions (no salt, 1 M NaCl (Na), 1 M NaCl with 5 mM $CaCl_2$ (Na/Ca)) (**Fig 5B**). The salt tolerance of knockout mutants that are part of the proposed Ca-dependent mechanism should not improve even with the addition of $Ca^{2+}$; in other words, the relative fitness should be lower under Na/Ca conditions than in Na conditions (indicated by the blue dots in **Fig 5C**). We identified 296 genes with lower relative fitness in the Na/Ca environment compared to Na (FDR $\leq$ 0.05, $\Delta Z \leq -1$) and found that these genes were enriched in the GO terms "mitochondria-related genes" and "ribosome" (**Fig 5D** and **S5 Table**).

We thus focused on the knockout strains of mitochondrial (Mito) genes. We examined the salt tolerance of the knockout mutants of the Mito genes and observed an interesting phenomenon; one group of Mito genes formed a distinct group with improved salt tolerance ($Z_{Na}-Z_{YPD} \geq 1$, Group I Mito. genes, **Fig 5E**). They appeared to be salt tolerant, but further investigation revealed that this was not the case. The reasons for this can be explained as follows. (1) Under non-stress conditions, these mutants have poor relative fitness ($Z_{YPD}$) due to their reduced proliferation compared to other mutants (**Figs 5F and S13A**, YPD). (2) Conversely, under salt stress, these mutants do not change their already impaired proliferation, while most other mutants show reduced proliferation. Their relative fitness ($Z_{Na}$) is thus the same as the other mutations (**Figs 5F and S13A**, Na). (3) The change in relative fitness between $Z_{Na}$ and $Z_{YPD}$ makes these mutants appear salt tolerant. However, $Ca^{2+}$ addition reduced their relative fitness ($Z_{NaCa}$) because they did not respond to $Ca^{2+}$, whereas the other mutants did (**Figs 5F and S13A**, Na/Ca, and **Fig 5H**). In summary, the knockout mutants of the Group I Mito genes grow slowly under non-stress conditions, but their growth is unchanged under salt stress, and they do not respond to $Ca^{2+}$ addition. Therefore, their normal function should be attacked by salt stress and restored by adding $Ca^{2+}$. We also observed that certain mitochondrial genes (referred to as Group II mito. genes, identified as $Z_{Na}-Z_{YPD} \leq -1$) displayed an

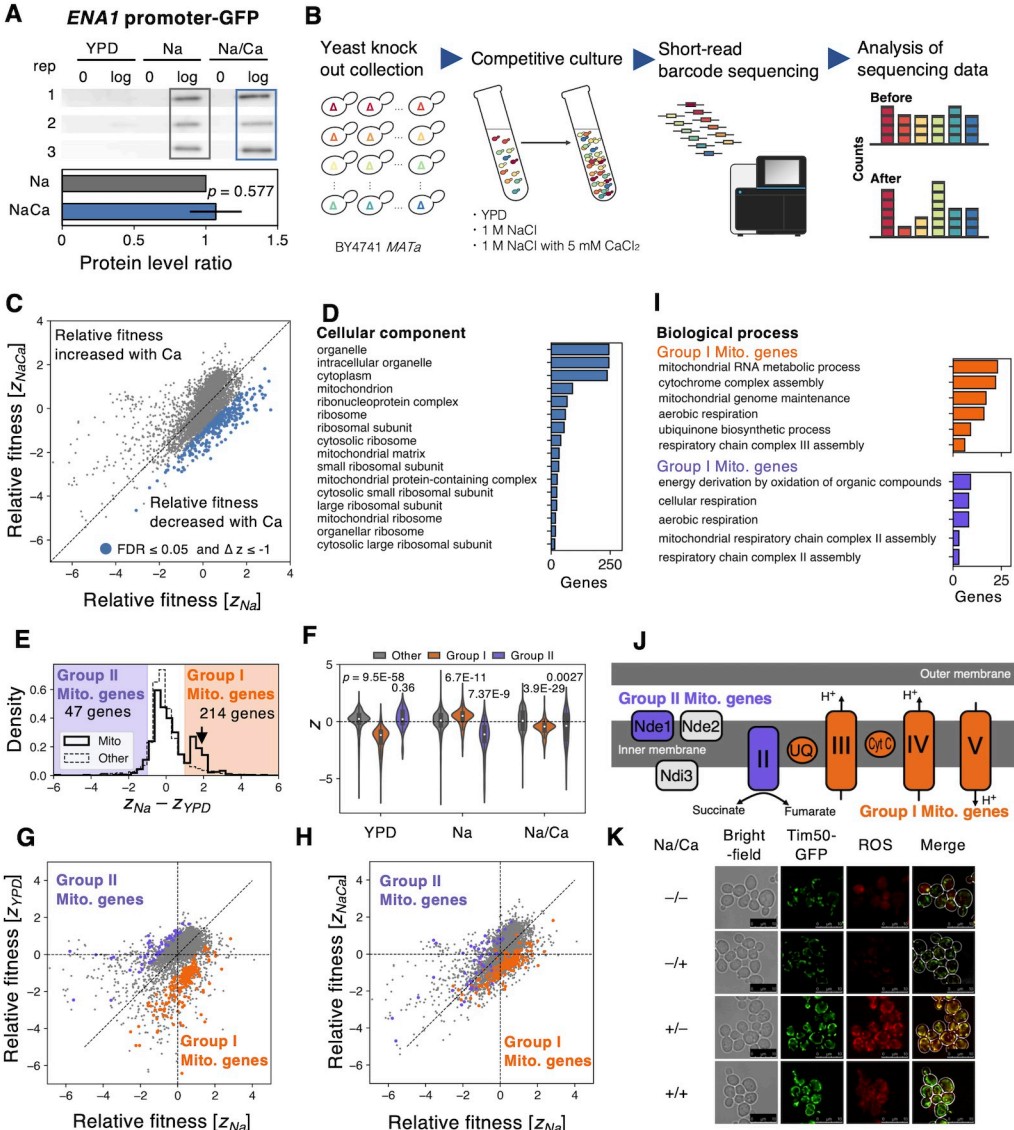

**Fig 5. Mitochondria appear to be a key target in the enhancement of salt tolerance by adding calcium. A)** Expression of *ENA1* under salt stress is not enhanced by CaCl₂ addition. The *ENA1* promoter activity was detected by Western blotting of EGFP under the control of the *ENA1* promoter under three conditions: YPD, 1 M NaCl (Na), and 1 M NaCl with 5 mM CaCl₂/YPD (Na/Ca). The lower panel shows the EGFP level in Na/Ca relative to Na during the logarithmic growth phase. The lower panel shows the EGFP level in Na/Ca relative to Na during the logarithmic growth phase. The error bar indicates the SD of relative values (*n* = 3). The *p*-value was calculated using Welch's t-test. **B)** A scheme of systematic analysis for relative fitness of gene knockouts. The detail is explained in the text. **C)** Comparing relative knockouts' fitness (*Z*) between Na and Na/Ca. The blue cycles indicate knockouts with reduced fitness (FDR ≤ 0.05 and *ΔZ* ≤ 1, Welch's t-test, and the Benjamini-Hochberg correction, *n* = 3). **D)** Enriched gene ontology (GO) terms in "cellular component" in the 296 knockouts with reduced fitness under Na/Ca (*p* ≤ 0.05, Holm-Bonferroni correction). The bar plot shows the number of genes with indicated GO terms. Other categories of enriched GO terms are shown in **S5 Table**. **E)** The distribution of fitness was corrected by YPD ($Z_{Na}$–$Z_{YPD}$). The solid and the dashed line indicate mitochondria (Mito) genes and the other genes, respectively. The orange area represents Group I Mito. genes ($Z_{Na}$–$Z_{YPD}$ ≥ 1), and the purple area means Group II Mito. genes ($Z_{Na}$–$Z_{YPD}$ ≤ –1). **F)** The distribution of relative knockouts' fitness of Group I (orange), Group II (purple), and the others (grey, 4,052 genes) under each condition. The *p*-values are from Welch's t-test by comparison with Other. **G-H)** Comparisons of relative knockouts' fitness between Na and YPD (**G**) and Na and Na/Ca (**H**) are shown. The purple and orange cycles indicate the knockouts belonging to Group I and Group II, respectively. The vertical and horizontal dashed lines indicate *Z* = 0. **I)** Enriched gene ontology (GO) terms in "biological function" of the knockouts belonging to Group I (upper, orange) and Group II (bottom, purple) (*p* ≤ 0.05, Holm-Bonferroni correction). A complete set of enriched GO terms can be found in **S6 Table**. **J)** The Group I and Group II Mito. genes have separate functions in the mitochondrial respiratory chain.

The complexes or proteins within Group I and Group II Mito. genes are colored orange and purple, respectively. **K)** Microscopic images of the cells with mitochondria and their reactive oxygen species (ROS) level under four conditions. Plus or minus of "Na" indicate YPD with or without 1 M NaCl, and plus or minus of "Ca" indicate with or without 5 mM CaCl$_2$. The green color shows the mitochondria inner membrane observed with Tim50-GFP. The red color indicates the mitochondrial ROS level stained by MitoTracker Red CM-H2Xros.

opposing response to Group I mito. genes (**Fig 5E-5H**). Under non-stress conditions, the growth of Group II genes is normal, but it is significantly reduced under salt stress, and they tend to respond positively to the addition of Ca$^{2+}$ (**Figs 5F** and **S13A**). This suggests that their normal function is to protect the cells from salt stress, but when Ca$^{2+}$ levels are adequate, this protective function is not required.

We then conducted a thorough examination of the Group I and II Mito genes. It was found that the molecular functions of the proteins encoded by these genes were distinct and specific to the segments before Complex II and after ubiquinone (UQ) of the mitochondrial respiratory chain, respectively (**Fig 5I and 5J**). From this, we hypothesized that salt stress leads to dysfunction in the respiratory chain (likely downstream of UQ) which is reversed by the addition of Ca$^{2+}$. To test this hypothesis, we observed mitochondria and reactive oxygen species (ROS) under salt stress with and without Ca$^{2+}$ using a fluorescence microscope (**Fig 5K**). As a result, we found that mitochondria were more developed and generated more ROS under salt stress, while the addition of Ca$^{2+}$ maintained mitochondrial development but suppressed ROS generation. Together, these findings led us to conclude that our proposed Ca$^{2+}$-dependent salt tolerance mechanism is related to mitochondrial function. Because of the high energy demand under salt stress [50], yeast cells likely require more productive mitochondrial function. This in turn generates high concentrations of ROS, causing growth defects, which may be suppressed by the addition of Ca$^{2+}$.

To further confirm the negative impact of salt stress on mitochondria, we analyzed transcriptome changes (RNA-seq) in the vector control and *CMD1-oe*, *ECM27-oe*, and *GDT1-oe* under salt stress conditions. The gene groups whose expression levels were significantly upregulated under salt stress conditions included "response to oxidative stress (GO:0006979)" and "arginine biosynthetic process (GO:0006526)" (**S14A and S14B Fig**). Additionally, we found that arginine uptake was significantly reduced by 20% under salt stress conditions (**S14C Fig**). Given that arginine synthesis is closely linked to mitochondrial function [51] (**S14D Fig**), we speculated that the elevated expression of these genes might be due to mitochondrial dysfunction under salt stress, which results in a reduction in arginine synthesis. This hypothesis was supported by the observation that the expression of these groups of arginine synthesis genes was commonly suppressed by *CMD1-oe*, *ECM27-oe*, and *GDT1-oe* (**S14E Fig**). These results align with our hypothesis that mitochondrial function is impaired under salt stress but is restored when calcium is supplied.

## Enhanced mitochondrial function can confer salt tolerance only when sufficient calcium is supplied

Why have no mitochondria-related genes been identified as GOFAs for salt tolerance, despite the critical role of enhanced mitochondrial function? It is possible that sufficient levels of Ca$^{2+}$ are required to prevent "mitochondrial runaway" under salt stress, as shown in **Fig 5K**. If this is the case, mitochondria-associated GOFAs should be identified specifically under salt stress conditions with added calcium. To test this hypothesis, we conducted overexpression profiling under 1 M NaCl with 5 mM CaCl$_2$ using Pool_a and Pool_b of BY4741 (**Fig 6A**). Interestingly, none of *CMD1*, *GDT1*, and *ECM27* were identified as hits upon the addition of calcium in any

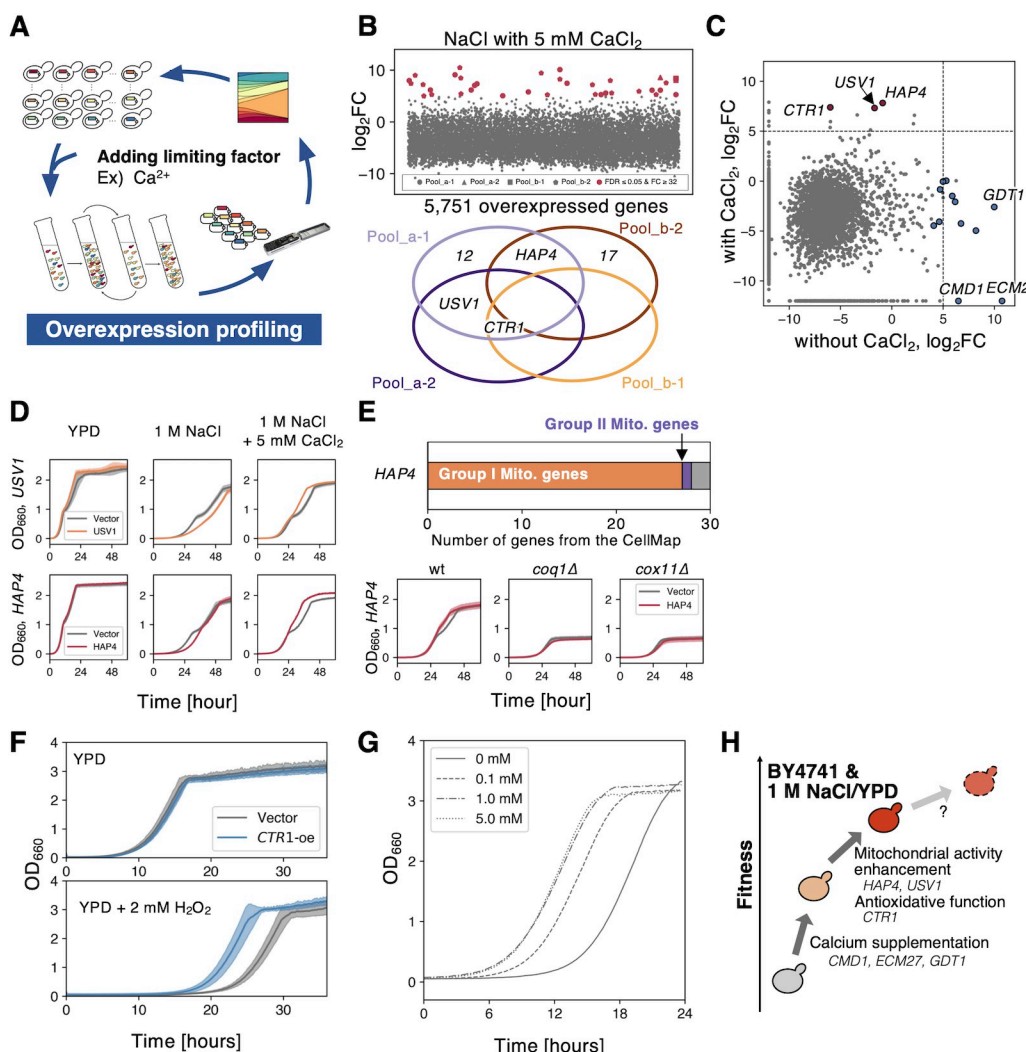

**Fig 6. Enhancing mitochondrial function can confer salt tolerance only when enough calcium is supplied. A)** Repeated isolation of GOFAs after adding the limiting factor CaCl₂ under salt stress. **B)** Fold change of plasmid occupancy after the 80 generations-cultivation of BY4741 overexpression library under 1 M NaCl with added 5 mM CaCl₂. (upper). Hit genes in each replicate under well-studied stresses (lower, FDR ≤ 0.05 and FC ≥ 2⁵). The log₂FC is plotted along the y-axis as a function of the 5,751 overexpressed genes ordered by ORF names. Hits are summarized in S7 Table. **C)** Compare the mean fold changes of plasmid occupancies with and without CaCl₂ addition. The colored circles indicate GOFAs, with (red) and without (blue) CaCl₂ addition. The dashed lines indicate the threshold of GOFAs as log₂FC ≥ 5. No values were replaced by -12. **D)** Growth curves of *USV1-oe* (upper, orange) and *HAP4-oe* (lower, red) under three conditions: YPD, 1 M NaCl, 1 M NaCl with 5 mM CaCl₂. The grey line shows the empty vector control. The filled areas indicate the standard deviation (*n* = 3). **E)** The upper panel: Most genes harboring genetic interaction with *HAP4* belong to the Group I Mito. genes (PCC ≥ 0.2, data obtained from The CellMap [54]). The lower panel: Growth curves of *HAP4-oe* in the deletion mutant of *COQ1* and *COX11*, the Group I Mito. genes harboring the genetic interaction with *HAP4* are shown. The filled areas indicate the SD (*n* = 3). **F)** *CTR1-oe* increased the growth of BY4741 under oxidative stress. The upper and lower panels indicate growth in YPD and 2 mM H₂O₂/YPD. The filled area shows the SD (*n* = 3). **G)** The addition of CuSO₄ increased the growth of BY4741 under 2 mM H₂O₂. Growth curves of cells with indicated CuSO₄ are shown. **H)** An illustration showing the mechanism of fitness increase of BY4741 under 1 M NaCl/YPD by fulfilling its requirements through GOFAs is shown.

replicate, further supporting the idea that these genes complement the Ca²⁺ requirement. On the other hand, a completely different group of genes (*CTR1*, *HAP4*, and *USV1*) were identified as hits shared in duplicates (hit in 2/4 *p* = 0.0001, hit in 3/4 *p* < 0.0001, and hit in 4/4

$p < 0.0001$, **Fig 6B and 6C**). *HAP4* and *USV1* have been previously reported as transcription factors for mitochondrial respiratory genes [52,53]. Of the 30 genes with genetic interactions with *HAP4* [54], 27 (PCC $\geq$ 0.2) belong to Group I Mito genes (**Fig 6E**), which strongly suggests a functional relationship between mitochondrial respiration and *HAP4*.

We confirmed that the overexpression of *USV1* and *HAP4* promoted growth under salt stress only when $Ca^{2+}$ was supplied (**Fig 6D**). Therefore, they are considered GOFAs under this condition. Interestingly, overexpressing these genes without $Ca^{2+}$ delayed growth under salt stress (**Fig 6D**), which supports the idea that enhanced mitochondrial function under salt stress without $Ca^{2+}$ is detrimental. Deletion mutants of Group I Mito. genes such as *COQ1* and *COX11* did not show an advantage with *HAP4* overexpression (**Fig 6E**), possibly because *HAP4* functions upstream of these genes. These results strongly support the idea that enhanced mitochondrial activity can confer salt tolerance only when sufficient calcium is supplied (**Fig 6H**), as reflected by the GOFAs identified under different conditions.

Finally, we focused on *CTR1*. *CTR1*, encoding a copper importer [55], and which was isolated as the GOFA under oxidative stress and calcium-supplemented salt stress (**Figs 1H and 6B**). *CTR1-oe* or adding $CuSO_4$ suppressed growth defects under oxidative stress (**Fig 6F and 6G**). Furthermore, instead of *CTR1*, the catalase genes *CTT1* and *CTA1* became GOFAs under oxidative stress supplied with 1 mM $CuSO_4$ [56,57] (**S15 Fig**). These findings indicate that copper is a critical limitation for oxidative stress and that adequate antioxidant function is necessary even when calcium is supplied under salt stress (**Fig 6H**).

## Discussion

In this study, we aimed to gain deeper insight into the mechanisms of stress resistance via overexpression. To do this, we developed an experimental system "overexpression profiling" to systematically isolate genes whose overexpression is functionally adaptive (GOFAs). We noticed that there were some technical limitations or biases when trying to isolate true GOFAs using our approach. 1) The competitive culture potentially accumulates genomic mutations adaptive to the stress, as observed in **Fig 2G**. 2) Isolation of genes was influenced by the initial pools (**S3 Fig**) because of the potential presence of preexisting genomic mutations or variations in the starting occupancy of cells with each plasmid. 3) Plasmid sequencing is biased towards certain insert lengths (**S4 Fig**). To minimize these biases and obtain confident GOFAs, we employed several strategies. We applied length correction (**S4 Fig**) to account for insert length bias in plasmid sequencing, used statistical thresholding (FDR $\leq$ 0.05 and $\log_2$FC $\geq$ 5, see Materials and Methods) to ensure the robustness of the results, and used biological replicates of at least triplicate to increase the confidence of our findings. We also used freshly prepared pools in later studies which further reduced the accumulation of mutations during storage. Our testing using freshly prepared transformants of GOFAs showed positive effects of the isolated genes (**Figs 2C, 3I, 4J, 6D, and 6F**), indicating that our approach effectively isolates GOFAs, and these results support the validity of our method.

We examined the characteristics of genes that become GOFAs under environmental stress. Our results revealed that GOFAs are genes that compensate for cellular deficiencies and that their adaptive function strongly depends on the genetic background and environment. For example, we found that GOFAs isolated under salt stress were associated with calcium homeostasis, and their adaptive function emerged from the lack of $Ca^{2+}$ in the medium (**Fig 2**). In fact, under $Ca^{2+}$-supplemented salt stress conditions, those adaptive functions were lost (**Fig 2F**), and other GOFAs than calcium homeostasis genes were identified (**Fig 6C**). In CEN.PK, genes such as $Na^+$ exporter *ENA1* and regulators of $K^+$ homeostasis (*SAT4* and *HAL5*) were identified as GOFAs under salt stress but not in BY4741 (**Fig 3H**). This difference in GOFAs

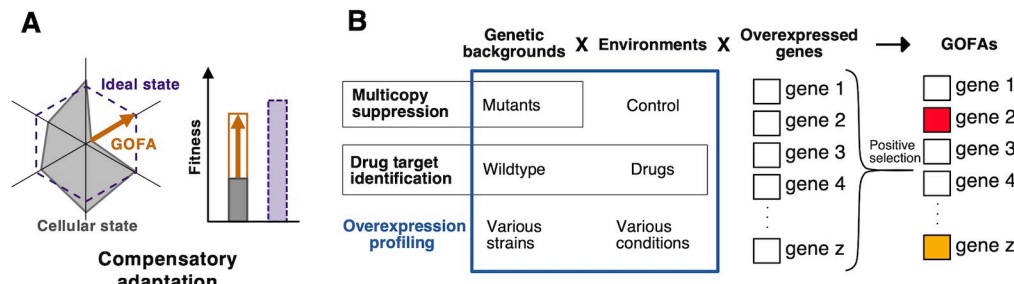

**Fig 7. Interpretation of overexpression profiling. A)** GOFAs manifest the potential for cellular stress tolerance (ideal state) due to compensating for cellular requirements. GOFA improves cellular fitness by supplementing elements (an arrow indicates) that are deficient compared to the ideal cellular state. **B)** Summary of overexpression profiling. Traditional multi-copy suppression or drug-targeted screening has focused on either the mutants or the drug. Overexpression profiling provides insight into cellular characteristics as a gene set by exploring GOFA in the context of environmental and genetic background interactions.

can be explained by the difference in $Ca^{2+}$ and $K^+$ requirements between BY4741 and CEN. PK. In fact, the adaptive effects of *ENA1-oe* and $K^+$ were more substantial in CEN.PK than in BY4741 (**Fig 3I**, **4E and 4H**). Based on these findings, we propose that GOFAs compensate for the missing elements needed for cells to reach maximum stress tolerance ("ideal state" in **Fig 7A**). In other words, examining GOFAs reveals the missing elements necessary to maximize cellular fitness within a given genetic background and environment.

It can be concluded that in the isolation of GOFAs, we observe a "compensatory adaptation/evolution" of cells to deficiencies. Besides being used as drug targets, genes that fall under GOFAs in this study are often explored as multicopy repressor genes [58,59], i.e., genes that suppress or compensate for the deleterious phenotype of a mutant by multicopy plasmids. In addition, in deleterious mutants subject to intense selection pressure, aneuploidy and consequent overexpression often occur to suppress or compensate for the harmful effects [58,60]. On the other hand, overexpression profiling provided a new means of observing the complementary adaptation/evolution of various strains against potential defects under multiple environments (**Fig 7B**).

The results suggest GOFAs compensate for cellular deficiencies to achieve a potential cellular stress response. In other words, GOFAs merely augment the existing stress response system. This approach does not fully explain why cells have evolved sophisticated stress responses or how they acquire new ones. This study focused only on genes that could explain the adaptive mechanisms. In fact, among GOFAs, there are also genes whose adaptive mechanisms cannot be easily described and "emerging genes" [40,61] such as *YBR196C-A* in Fig 2A. The mystery of the evolution of novel stress response mechanisms may be hidden in these genes. It may also be that the byproducts of compensatory adaptation to the environment appear as functional novelties, similar to the morphological novelties that occur in the compensatory evolution of gene loss [62].

Through the identification of GOFAs under salt stress using overexpression profiling, we found that calcium has a positive effect on long-term salt stress, distinct from the previously known short-term response to salt stress (**Figs 2 and S12**). Additionally, functional profiling of gene disruption mutants revealed that mitochondrial runaway might be suppressed by $Ca^{2+}$ (**Fig 5**). Overexpression of GOFAs (*HAP4* and *USV1*) identified under salt stress with supplied $Ca^{2+}$ seemed to enhance mitochondrial function, working positively for salt stress tolerance only under calcium-supplying conditions but negatively under calcium-limiting conditions (**Fig 6D**). This suggests that the primary function of calcium is to regulate mitochondrial

activity under salt stress. As shown in these experiments, the advantage of overexpression profiling is its ability to identify GOFAs quickly and efficiently in various strains and environmental conditions (Fig 7B). By using "overexpression profiling," we can uncover previously unexplored mechanisms of cellular adaptation.

## Materials and method

### Strains and plasmids

The strains and plasmids used in this study are listed in S1 Table.

### Medium and yeast transformation

The yeast culture and transformation were performed as previously described [63]. We used two types of medium: YPD and Synthetic Complete (SC) medium. YPD included 10 g/L Bacto Yeast extract (BD, USA), 20 g/L Bacto Peptone (Gibco, USA), and 20 g/L D-glucose. The SC medium included 6.7 g/L Yeast Nitrogen Base with Ammonium Sulfate (MP, USA), 0.65 g/L DO supplement-HisLeuUra (Clontech, USA), and 20 g/L D-glucose, or where appropriate, 20 mg/L Histidine, 8 mg/L Uracil, and 100 mg/L Leucine. The D-glucose solution was added to the medium after autoclaving. Milli-Q water (Merck, Germany) was used to condition the medium. In Fig 6D and 6E, YPD and 1 M NaCl/YPD were diluted four times with sterile water or 1 M NaCl solution. We used Shio (Shiojigyo, Japan) and Setonohonjio (Ajinomoto, Japan) as the table salt and crude salt representatives.

### Plasmid and strain construction

Plasmid and strain construction was performed as described previously [64]. The plasmids and strains were constructed by homologous recombination activity in yeast cells, following a previous report [65], and their plasmid construction was verified by Sanger sequencing.

### Exploring well-studied stress

Using the PubMed API provided by the National Institutes of Health (NIH) and the efetch of E-utilities, we obtained 308,970 and 20,460 articles (as of May 20th, 2022) by searching for "yeast" and "yeast stress" (including authors, titles, abstracts, year of publication, and journal title), respectively. Using the TF-IDF method, we extracted the keywords with the highest scores up to the 5th place from the abstracts obtained for "yeast stress" [66]. Among all keywords, keywords, including "stress" were extracted. Next, we determined whether each stress keyword appeared in the abstract of the articles obtained by "yeast stress." We used the NLTK library (version 3.6.7) for these analyses and a Python library for natural language processing [67].

### Growth rate assay

The target strains were inoculated into 4 ml of SC (–Ura or–HisUra) medium in test tubes and incubated overnight at 30°C as a pre-cultivation. Then, 25 μl of the pre-cultured medium was inoculated into 6 ml of the target medium in L-shaped tubes and cultured at 30°C (excluding heat stress), recording the optical density (OD) at 660 nm every 10 minutes with an ADVANTEC TVS062 (ADVANTEC, Japan) while shaking at 70 rpm. Growth rates [1/hour] were calculated from the recorded OD data as the reciprocal of the mean doubling time, which was the slope of log base 2 of $OD_{660}$ between 0.125 and 0.500 by linear approximation with scipy.optimize.curve_fit in the Python library. If the $OD_{660}$ did not exceed 0.125 48 hours after inoculation, we designated it as not detected (N.D). We used freshly prepared plasmid transformants

that had not been exposed to stress conditions to ensure that any observed growth enhancements were a result of the overexpression of specific genes rather than mutations that occurred during adaptation to stress.

## Overexpression profiling

As the first step of the overexpression profiling, we constructed three types of overexpression libraries. 1) BY4741-overexpression library: 5 μl of thawed gTOW6000 collection [21] was inoculated into sixty 96-well plates with 200 μl of SC–Ura medium and incubated at 30˚C for 48 hours. All cultured mediums were then pooled in sterile flasks and divided into 50 ml tubes. Finally, pooled libraries were added with DMSO to a final concentration of 7% v/v and stored at -80˚C. 2) CEN.PK2-1C and DBVPG6765-overexpression libraries: Each insert fragment of the gTOW6000 library was amplified by PCR using specific primer sets, as previously described [21]. The PCR reactions were performed in a 96-well format on a 50 μL scale using KODplus NEO (Toyobo) for 61 sets. All 96 reactions were pooled (4.8 mL) and mixed in equal volumes to create the mixed PCR products. The two plasmid fragments from pTOW40836 were amplified by PCR using two pairs of primers: 5'- GGATCCACTAGTTCTAGAGCGGC CG-3' and 5'- GCTCGTTACAGTCCGGTGCG -3', 5'- CTCGAGGGGGGGCCCGGTACC CAATTCGCCCTATA-3' and 5'- ACGAATGCACACGGTGTGGTGG -3'. The mixed PCR products and plasmid fragments were introduced into the target strain. Transformation protocols were performed according to the previous report [63]. To 6 ml of yeast with 1 $OD_{660}$ unit, 60 μl of PCR mix, 30 μl of plasmid fragments, 1,440 μl of 50 w/v % polyethylene glycol 4,000, 216 μl of 1 M LiOAc, and 144 μl of ssDNA were added and spread on 15 cm diameter SC–Ura agar medium. Incubated at 30˚C for 48 hours, scraped off the colonies on all agar plates, and added DMSO to final conc. 7 v/v%, and stored at -80˚C. Yeast cells were equivalent to 2 agar plates for DBVPG6765 and 5 agar plates for CEN.PK2-1C was transformed. 3) CEN. PK2-ENA1 co-overexpression library: The CEN.PK2-1C-overexpresson library and CEN. PK2-1D bearing pRS423nz2-ENA1 were mixed to be 1:1. Each used 2 OD units. Mixed strains were spread and cultured on an SC–HisUra agar plate (15 cm) for 48 hours at 30˚C. After incubation, to selectively reduce unmated yeasts, the colonies were scraped, and 2 OD units each were spread again on 5 plates of fresh SC–HisUra agar and incubated at 30˚C for 48 hours. Finally, the colonies were scraped off, and DMSO was added to reach the final conc. 7%, and stored at -80˚C.

The second step of the overexpression profiling involved competitive culture and passage. 1 ml of the overexpression library was inoculated into 5 ml of SC–Ura medium and incubated at 30˚C overnight with shaking. Then, 24 μl (1:250 dilution) of the pre-cultured medium was inoculated into 6 ml of the target medium in L-shaped tubes and cultured until the stationary phase, measuring the optical density with an ADVANTEC TVS062 (ADVANTEC, Japan). 24 μl (1/250 dilution) of the pre-cultured medium was passage into 6 ml of fresh medium in L-shaped tubes. The passage was repeated 1–10 times. For heat stress, the cultures were performed at 37˚C or 40˚C in a bio-shaker BR-43FL (TAITEC, Japan) set at 35˚C. The all-cultured medium was transferred into a 5 ml tube and centrifuged at 15,000 rpm for 1 min, its supernatant was removed, and 1 ml of 10% v/v DMSO water was added and suspended. The suspension was transferred to 1.5 ml tubes and stored at -80˚C. For the methotrexate experiment, competitive cultures were made in 500 ml Erlenmeyer flasks with 150 ml of medium, and 4.5 μl (about 1:33,000 dilution) of the culture was inoculated and passaged.

As the third step of the overexpression profiling, plasmid preparation from the competitively cultured yeast and long-read sequencing were performed as follows. 500 μl of the thawed sample was transferred to a new 1.5 ml tube, and the remaining sample was re-stored at -80˚C.

The sample was centrifuged at 15,000 rpm for 1 min, and the supernatant was removed. The sample was then resuspended in 250 μl of Solution 1 (1 M sorbitol, 0.1 M $Na_2EDTA$ (pH 7.5), and 10 μg/ml RNase) and 5 μl of 10 units/μl Zymolyase–T100 (Nacalai tesque, Japan) and incubated for 30 minutes. 250 μl of Solution 2 (0.2 M NaOH and 1% w/v SDS) was added to the suspension and mixed. Then, 250 μl of Solution 3 (3 M potassium acetate and 2 M acetic acid) was added to the suspension and vortexed. The suspension was centrifuged for 10 minutes at 15,000 rpm to precipitate the insoluble material. The supernatant was added to a spin column (QIAprep Spin Miniprep Columns, Qiagen, Germany) and centrifuged at 13,000 rpm for 1 minute. The column-through effluent was removed, and 750 μl of wash buffer (QIAprep Spin Miniprep Kit, Qiagen, Germany) was added and centrifuged for 1 minute. The column-through effluent was removed, and the empty column was centrifuged at 13,000 rpm for 1 minute to dry the column.

Long-read sequencing for plasmid inserts was performed as follows. The sequencing library preparation was performed according to the manufacturer's instructions, "Four-primer PCR protocol," using SQK-PBK-004 (Oxford Nanopore Technologies, UK). 25 ng of purified plasmid was used as each sample, and PCR reactions were performed with half of the defined protocol and the own designed primers 5'-TTTCTGTTGGTGCTGATATTGCggcgaaaggggggatg tgctg-3' and 5'-ACTTGCCTGTCGCTCTATCTTCggaaagcgggcagtgagcgc-3'. The libraries were sequenced using GridION or MinION and MinIT (Oxford Nanopore Technologies, UK) with the flow cell MinION R9.4.1. 6–12 samples per flow cell were analyzed in multiplexing. Base-calling and demultiplexing were performed using MinKNOW (Oxford Nanopore Technologies, UK) with guppy in high-throughput mode.

As the last step of overexpression profiling, analysis of sequence data and identification of GOFAs were performed following. Sequence data (fastq format) was aligned to a reference genome sequence file (R64-1-1) of budding yeast S288C using minimap2 (2.24) [68] to output an alignment file sam format). Next, the alignment file was reformatted and sorted using "view -Sb" and "sort" in Samtools (1.15) [69] to obtain a bam format file. Then, Bedtools (2.30.0) [70] with "bamtobed" converted the bam format file to a bed format file. The aligned reads on the gTOW6000 insert locus were extracted using "bedtools intersect" with an option "-F 0.5". The read counts on insert locus were counted by "bedtools coverage." Subsequent analyses were performed using python (3.8.12) with NumPy (1.21.2) and pandas (1.4.1) and visualized using IGV[71]. Reads for each insert were converted to reads per million (RPM). The fold change of plasmid occupancies was calculated according to the following equation,

$$FC_{n,i} = RPM_{n,i}/RPM_{0,i}$$

where $RPM_{0,i}$ is the ratio of insert i in the pool before competitive passages and $RPM_{n,i}$ is the ratio of insert i after n passages. In this study, genes with a larger fold change than 32 ($2^5$) and the false discovery rate (FDR) $\leq 0.05$ were considered hits. FDR was calculated for each count data in each replicate by chi-square test and Benjamini-Hochberg method [72]. The chi-square test used the ratio of plasmid appearance before and after competitive culture and the count data as parameters. The diversity of plasmids was evaluated using the Gini-Simpson index, calculated following [73,74]. The permutation test was performed using Python scripts to assess the significance of the observed overlap of genes in replicates by comparing the observed overlap to the distribution of overlaps obtained from random permutations of the data. The main steps in the analysis are as follows: a) Input the gene hits for each of the three replicates, along with the total number of genes. b) Define a function to calculate the overlap between replicates for shared gene hits. c) Define a function to randomly permute the gene hits in the replicates while preserving the number of hits in each replicate and calculate the overlap for the

permuted data. d) Perform the permutation test by comparing the observed overlap to the distribution of overlaps obtained from random permutations (10,000 in this case) of the gene hits in the replicates. e) Calculate a p-value for each category (number of replicates with shared gene hits) by counting the proportion of permutations with an overlap equal to or greater than the observed overlap. The raw data were available in the DNA Data Bank of Japan (accession number: DRA015709).

## Aequorin assay

In order to construct the plasmid pEVP11/AEQ-HIS3, the *LEU2* locus of the plasmid pEVP11/AEQ was replaced with the *HIS3* gene. This was achieved by using PCR to amplify the *HIS3* gene fragments from the template plasmid pRS413 using the primers 5'-GGCCGA GCGGTCTAAGGCGCgtttcggtgatgacggtgaa-3' and 5'-GCGCTGGGTAAGGATGATGCgcc gatttcggcctattggt-3'. Additionally, the fragments of pEVP11/AEQ without the *LEU2* locus were amplified from pEVP11/AEQ using the primers 5'-ttcaccgtcatcaccgaaacGCGCCTTAGACC GCTCGGCC-3' and 5'-accaataggccgaaatcggcGCATCATCCTTACCCAGCGC-3'. The resulting *HIS3* and pEVP11/AEQ fragments were then used to construct pEVP11/AEQ-HIS3. This plasmid was then introduced into each overexpressing strain using transformation protocols according to Amberg 2005. The target strains were inoculated into 4 ml of SC (–Ura or–HisUra) medium in test tubes and incubated overnight at 30˚C as pre-cultivation. Afterward, 200 µl of pre-cultured medium were inoculated into 5 ml YPD medium and cultured until $OD_{660}$ reached 1.0. One $OD_{660}$ unit was then aliquoted into a 1.5 ml tube and centrifuged, and its supernatant was removed. The pellet was resuspended with 50 µl YPD, including 5 mM Coelenterazine H (Wako, Japan), and stood in the dark at room temperature for one hour. After centrifugation and removing the supernatant with Coelenterazine H, the pellet was washed with fresh YPD, suspended in 75 µl of YPD medium, and then applied to 96 well plates. Luminescence intensity was measured using a microplate reader MTP-880Lab (CORONA, Japan). Fluorescence intensity was measured for 50 seconds at 5-second intervals as a baseline. Then, 25 µl of 4 M NaCl solution was added to the well by the automatic dispenser DP-50N (CORONA, Japan). The plate was agitated for 5 seconds, and the fluorescence intensity was measured every 5 seconds for 10 minutes.

## Measurement of mineral concentration in the medium

We utilized LAQUAtwin ionometers (HORIBA, Japan) to measure the mineral concentrations in the medium used. The measurements were performed according to the manufacturer's instructions. 500 µl of each medium was placed on the sensor of the ionometer, and the concentrations of sodium (Na-11), potassium (K-11), and calcium (Ca-11) were measured.

## Laboratory evolutionary experiment

The culture and passages followed the "overexpression profiling" described above. 1 ml of BY4741 bearing pTOWug2836 as vector control was inoculated into 5 ml SC–Ura medium and incubated at 30˚C overnight with shaking. Ten passages were cultured in YPD medium containing 1 M NaCl.

## Genome preparation

The genome of pooled cultured strains was extracted according to a previous report 71. 500 µl of Solution 1 (1 M sorbitol, 0.1 M $Na_2EDTA$ (pH 7.5), and 10 µg/ml RNase) and 5 µl of 10 units/µl Zymolyase solution were added to the pellet of 5 $OD_{660}$ units of cultured yeast. The

mixture was then suspended and incubated at 37˚C for 30 minutes. After centrifugation and removal of the supernatant, 250 μl of buffer (20 mM Na2EDTA and 50 mM Tris-Cl (pH 7.4)) and 25 μl of 10% SDS were added, and the mixture was incubated at 65˚C for 30 minutes. 100 μl of 5 M potassium acetate was added to the sample, which was cooled on ice for 30 minutes. After centrifugation, the supernatant was transferred to a new 1.5 ml tube. 400 ml of isopropanol was added to the supernatant and placed at room temperature for 5 minutes. After centrifugation, the pellet was rinsed with 70 v/v% ethanol. 50 μl of sterile water was added to the pellet to extract the genome. The extracted genome solution was stained with a DNA staining reagent (Qubit 1X dsDNA HS Assay Kit, ThermoFisher), and the plasmid concentration was measured with a Fluorometer (Qubit4, ThermoFisher).

## Genome sequencing and variants calling

Genome quality check and resequencing were outsourced to Macrogen Japan (Japan). Library preparation was performed using TrueSeq DNA PCR Free Kit (Illumina, USA), and sequencing was performed using NovaSeq 6000 (Illumina, USA) under 150 bp paired-end conditions to obtain sequence data in fastq format files. Sequence data were aligned and mapped to a reference genome sequence file (R64-1-1) of budding yeast S288C using BWA (0.7.17) [75]. Next, the alignment file (SAM format) was converted to a bam format file and sorted using Samtools (1.15). Variants for each sample were called and performed using Bamtools (1.15) with "mpileup"[76] with "call" and filtered using vcfutils.pl varFilter (default parameters). Variants were annotated by snpEff (4.1) [76,77] using R64-1-1.86. A comparison of An and Ev variations was performed by bcftools isec. Called variants were checked manually using IGV (2.8.10) and validated by chi-square test for base composition between An and Ev.

The threshold for validation was set at an FDR of 0.05 or less, corrected by the Benjamini-Hochberg method[72]. The raw data were available in the DNA Data Bank of Japan (accession number: DRA014470).

## GFP western blot analysis

The detection of GFP was performed by Western blot as previously described [78]. ENA1-GFP cells were grown in YPD, 1 M NaCl/YPD, and 1 M NaCl/YPD with 5 mM $CaCl_2$. The cells were harvested at the log phase ($OD_{660}$ = 1.0) and treated with 1 ml of 0.2 M NaOH. 50 μl of 1xNuPAGE LDS sample buffer (Invitrogen, USA) was added, and the mixture was heated at 70˚C for 10 minutes. The protein lysate was labeled with Ezlabel FluoroNeo (ATTO, Japan) and separated by polyacrylamide gel electrophoresis on 4–12% NuPAGE 4%–12% Bis-Tris Gel (Invitrogen, USA). The separated proteins were transferred onto a PVDF membrane (Invitrogen, USA) using the iBlot (Invitrogen, USA). GFP was probed using an anti-GFP antibody (Roche) (1:1,000) and a peroxidase-conjugated secondary antibody (Nichirei Biosciences, Japan) (1:1,000) followed by SuperSignal West Femto Maximum Sensitivity Substrate (Thermo Fisher Scientific, USA). The band intensity was detected and measured using the LAS-4000 image analyzer (Fujifilm, Japan) and quantified using ImageJ (1.53k).

## Genetic profiling using yeast gene knockout collection

96-well plates were dispensed with 200 μl of YPD and inoculated with 5 μl of thawed Yeast Knockout Out Haploid MAT-a Collection [79],. The plates were incubated at 30˚C for 48 hours. All culture strains were mixed in sterile flasks and divided into 50 ml tubes. The culture was added DMSO to a final concentration of 7% v/v and stored at -80˚C; we named it KO library. 1 ml of KO library was inoculated into 5 ml YPD medium and incubated at 30˚C overnight with shaking. 6 ml of medium was dispensed into an L-shaped tube, and 24 μl (1/250

dilution) of the pre-culture was inoculated and incubated for a fixed time until steady-state while measuring optical density with an ADVANTEC TVS062. After a particular time, 6 ml of fresh medium was dispensed into another L-shaped tube, and the culture was passaged 24 μl (1/250 dilution) of the culture. The passage was repeated two times. The genome of harvested cells was extracted. Strain-specific DNA barcodes were amplified using multiplex primers and a common U2 primer. PCR conditions were set as follows: 5 min at 98˚C for initial denaturation, 30 cycles of 30 sec at 98˚C, 30 sec at 55˚C, 45 sec at 72˚C, and a final extension time of 10 min at 72˚C. PCR products were purified from 2% agarose gels using a Geneclean III kit, quantified using a Kapa qPCR kit, and sequenced with an Illumina HiSeq 2500 machine. Sequence analysis was performed on the second passages and the pre-culture pool. Each experiment was performed in biological triplicates for all conditions. We denoted relative fitness in terms of $Z$-scores, which was the standard normalized distribution of fold change between RPM of barcodes before and after cultivation. FDR and the $Z$-score between conditions were calculated using Welch's t-test and the Benjamini-Hochberg correction [72,80]. Gene ontology enrichment analysis was performed using the Gene Lists function on Saccharomyces Genome Database (SGD) website (www.yeastgenome.org/).

## Microscopic observation

Microscopic observation was performed as described previously [81]. TIM50-GFP were cultured in YPD, YPD with 5 mM $CaCl_2$, 1 M NaCl/YPD, and 1 M NaCl/YPD with 5 mM $CaCl_2$. Cells were harvested at the log phase ($OD_{660}$ = 1.0), and 1 μl of the suspension cell was mixed with 2 μl of YPD on a glass slide. Images were obtained and processed using the DMI6000 B microscope and Leica Application Suite X software (Leica Microsystems, Germany). The GFP fluorescence was observed using the GFP filter cube (Leica cat. # 11513899). Mitochondria were stained with 100 nM of MitoTracker Red CM-H2Xros (M7513, Thermo Fisher Scientific, USA) for 30 min and then washed with 0.5 ml of YPD. The cells were then observed using RFP filter cubes (Leica cat. # 11513894).

## RNAseq

RNA sequencing was performed as previously described [82]. The four strains: *CMD1-oe*, *ECM27-oe*, *GDT1-oe*, and vector control, were pre-cultured in SC–Ura at 30˚C overnight and then cultured in YPD or YPD with 1M NaCl medium. The cells were harvested at the log growth phase ($OD_{660}$ = 1.0). Purified RNA was quality-checked using BioAnalyzer (Agilent, USA) or MultiNA (Shimazu, Japan), and the concentration was measured using Qubit (Thermo Fisher Scientific, USA). Purified RNA was stored at -80˚C until further experiments. A cDNA library was prepared using the TrueSeq Stranded Total RNA kit (Illumina, USA) and following half of the protocol of the TrueSeq RNA library prep kit. 4 μg of the library was prepared by adding 1 μl of 142.8x diluted ERCC RNA Spike-in mix (ThermoFisher, USA) to 4 μg of total RNA. The libraries were quality checked using an Agilent 2100 BioAnalyzer (Agilent, USA), concentrations were measured on a Real-Time PCR system (ThermoFisher, USA), and the libraries were pooled. cDNA library sequencing was performed using pair-end sequencing on an Illumina NextSeq 550 (Illumina, USA). Three biological replicates were analyzed for all strains. The sequences were checked for sequence quality using FastP (0.23.2) [83] and then aligned using Hisat2 (2.2.1) [84]. The aligned data were formatted into bam files using Samtools (1.15) [69]. Finally, expression level variation analysis was performed using EdgeR (3.40.0) [69,85]. The raw data is available in the DNA Data Bank of Japan (accession number: DRA014472). GO enrichment analysis was performed using the Gene Lists function on the SGD website (www.yeastgenome.org/).

## Arginine uptake assay

BY4741 wild-type and *can1Δ* cells were cultured in YPD or YPD with 1 M NaCl medium, harvested at the log growth phase ($OD_{660}$ = 1.0) and suspended in 4 ml of YPD or YPD with 1 M NaCl medium at a density of 2.5 x $10^8$ cells/ml, respectively. The arginine uptake reaction was initiated by adding 1.0 ml YPD or YPD with 1 M NaCl medium containing [U-14C] arginine at a final radioactivity level of 0.518 kBq/ml. The cells were then incubated for 60 min at 30˚C. 0.5 ml aliquots of the cell suspension were withdrawn and filtered on cellulose acetate membrane filters (0.45 μm) and washed with cold 10 mM HEPES (pH6.4). The radioactivity of the recovered cells was measured using a liquid scintillation counter, and arginine uptake was calculated by subtracting the radioactivity at 0 min from that at 60 min. To normalize the arginine uptake, protein content in cells mixed with YPD not containing radiolabeled arginine and collected at 0 min and 60 min were measured using the Lowry method [86].

## Quantification and statistical analysis

Information on statistical analysis and biological replicates is included in figure legends. The significance level was set at 0.05.

## Supporting information

**S1 Fig. Coverage of pooled libraries.**
(PDF)

**S2 Fig. Growth of yeast cells under three most-studied stresses.**
(PDF)

**S3 Fig. Reproducibility of overexpression profiling.**
(PDF)

**S4 Fig. Correction of frequency bias by insert length.**
(PDF)

**S5 Fig. Behaviors of dosage-sensitive genes and stress-induced genes in overexpression profiling.**
(PDF)

**S6 Fig. Effect of overexpression of *NCS2* and *NCS6* at high temperature.**
(PDF)

**S7 Fig. The addition of $Ca^{2+}$ increased the growth rates of various strains under salt stress but not without salt stress.**
(PDF)

**S8 Fig. Supplement to overexpression profiling of CEN.PK and DBVPG6765.**
(PDF)

**S9 Fig. Schematic diagram of the *PMR2* locus in BY4741, DBVPG6765, and CEN.PK.**
(PDF)

**S10 Fig. The quality check of the CEN.PK2-*ENA1* co-overexpression library.**
(PDF)

**S11 Fig. Effect of different salt sources on the growth of yeast.**
(PDF)

**S12 Fig. The effects of calcium addition alone cannot be explained by short-term stress response enhancement.**
(PDF)

**S13 Fig. Relative fitness distribution when the growth rates in each condition were subtracted from Fig 5F.**
(PDF)

**S14 Fig. Transcriptome analysis of cells under salt stress obtained by RNAseq analysis.**
(PDF)

**S15 Fig. GOFAs enriched under oxidative stress propose $Cu^{2+}$ limitation in the culture medium.**
(PDF)

**S1 Table. Used strains and plasmids in this study**
(XLSX)

**S2 Table. Hit genes under well-studied stress**
(XLSX)

**S3 Table. Variants calling of lineage1 and linage 2.**
(XLSX)

**S4 Table. Hit genes in various strains under salt stress.**
(XLSX)

**S5 Table. GO terms enriched in 296 genes with lower relative fitness in the Na/Ca environment compared to Na**
(XLSX)

**S6 Table. GO terms enriched in Group I and II mito genes.**
(XLSX)

**S7 Table. Hit genes in Ca-supplied salt stress and Cu-supplied oxidative stress.**
(XLSX)

**S8 Table. DEGs in overexpression strains under salt stress.**
(XLSX)

**S9 Table. Permutation test to evaluate the significance of the observed overlaps.**
(XLSX)

**S1 Data. Raw data for figures.**
(ZIP)

## Acknowledgments

We thank the members of the Moriya Laboratory (Okayama University) for their helpful discussions. We thank Nobuyuki Uozumi and Patrick Masson for sharing pEVP11/AEQ, Michael Knop for the concept inspiration, Luis Alberto Vega Isuhuaylas for chemical genomics analytical support, Hiroaki Mano, Yuuki Kobayashi, Shoko Ohi, Mika Ikeda, and Wen Xin Xuan for high-throughput sequencing support.

## Author Contributions

**Conceptualization:** Yuichi Eguchi, Hisao Moriya.

**Data curation:** Nozomu Saeki, Chie Yamamoto, Yuichi Eguchi, Takayuki Sekito, Mami Yoshimura, Yoko Yashiroda.

**Funding acquisition:** Hisao Moriya.

**Methodology:** Nozomu Saeki, Shuji Shigenobu, Yoko Yashiroda, Charles Boone.

**Software:** Nozomu Saeki.

**Supervision:** Shuji Shigenobu, Charles Boone, Hisao Moriya.

**Visualization:** Nozomu Saeki.

**Writing – original draft:** Nozomu Saeki, Hisao Moriya.

**Writing – review & editing:** Nozomu Saeki, Yoko Yashiroda, Hisao Moriya.

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
