## [Decision Letter · Decision Letter 0]

21 Nov 2022

Dear Dr Moriya,

Thank you very much for submitting your Research Article entitled 'Overexpression profiling reveals cellular requirements in the context of genetic backgrounds and environments' to PLOS Genetics.

The manuscript was fully evaluated at the editorial level and by independent peer reviewers. The reviewers appreciated the attention to an important problem, but raised some substantial concerns about the current manuscript. Based on the reviews, we will not be able to accept this version of the manuscript, but we would be willing to review a much-revised version. We cannot, of course, promise publication at that time.

If you decide to revise the manuscript for further consideration at PLOS Genetics, please aim to resubmit within the next 60 days, unless it will take extra time to address the concerns of the reviewers, in which case we would appreciate an expected resubmission date by email to plosgenetics@plos.org.

We are sorry that we cannot be more positive about your manuscript at this stage. Please do not hesitate to contact us if you have any concerns or questions.

Yours sincerely,

Joseph Schacherer, Ph.D.

Guest Editor

PLOS Genetics

Geraldine Butler

Section Editor

PLOS Genetics

The overall feeling of the 3 reviewers is that this study is of interest. However, the paper cannot be published in its current form and some issues have been raised by the reviewers. In particular, the reviewer 1 pointed out the lack of statistical analysis and this point needs to be addressed. The reviewer 2 highlighted the fact that a better control and evaluation of the potential impact of 'primary mutation' in the experiments should be proposed. And overall, the reviewers felt the introduction should be rewritten to better present the state of the art.

Reviewer's Responses to Questions

**Comments to the Authors:**

Reviewer #1: Saeki and colleagues performed overexpression profiling of a library of yeast genes in various stress conditions, identifying a variety of genes and pathways that allow improved growth when overexpressed. They found that the particular overexpression hits were dependent on strain background, and they also tested how changing the environmental conditions affected the hits that were returned.

Concerns:

- Many key experiments in the manuscript lack statistical analysis. In particular, the hits from the overexpression profiling were determined using an enrichment threshold following selection, but the authors generally did not calculate a false discovery rate associated to the threshold (the one exception being fig. 5D). The authors performed four replicate experiments, but it does not appear that they leveraged the replicates to determine statistical significance.

- The authors quantify the abundances of their overexpression constructs following selection as “degree of enrichment,” which I interpreted as a measure of fold-change following selection, so the threshold used by the authors, 10,000, seemed like quite an impressive bar to clear. But “degree of enrichment” is not actually defined in the Results section (I recommend doing so); in the Methods section it is described on line 618 as “the number of plasmids per million plasmids after [selection], assuming that the initial pool was completely uniform.” I would not call that a “degree of enrichment,” since it does not reflect fold-change, that is, enrichment. Probably a clearer description would be “normalized reads per million.” But why not use a simple fold-change measure?

- The manuscript seems to have an error when defining the “degree of enrichment” discussed above, which further confuses the matter. An equation is given in the Methods (line 616) for calculating this value, but a different equation is given in Extended Data Figure 3C; the two equations differ by a factor of 10^6. I suspect the equation in the Methods section is incorrect and the one in the Extended Data Figure is correct. I also recommend using consistent nomenclature; “RPM” can be used in both places, instead of “p,” which is only used in the Methods.

- The authors tested the overexpression library in additional strains. The Methods section describes the construction of this library as follows: “The mixed PCR products of S288C’s ORF and the two plasmid fragments from pTOW40836 were introduced into the target strain,” on line 685. I do not see any explanation of how this PCR was generated – what was the template? The primers? Was it a single PCR from some sort of DNA library, or separate PCRs that were pooled? Nor is there information on what the two plasmid fragments are. The Materials and Methods section needs to be detailed enough that a skilled independent investigator would be able to repeat the experiments with just the information provided. That is certainly not the case for this experiment, and I recommend the authors reconsider the entire Methods section with that standard in mind.

- The authors sequenced their control strains’ genomes following growth in salt stress, and mutations were identified in PMA1 and PMR1. (Section starting Line 652.) Were single colonies picked, or was the entire pool sequenced? Were these the only mutations identified?

- The authors provide p-values for luminescence intensity comparisons in figure 2D, but I could not tell what quantification of luminescence intensity was used in the statistical test, as the figures show values across a time window for each strain. Was the maximum value used? Or was it the area under the curve? Something else?

- The authors state that there is “high energy demand under salt stress” (line 370). If this statement is based on work in the literature, a citation is needed. If the statement derives from results in this manuscript, I did not understand which results showed the statement to be true, so the basis for the statement should be more clearly explained.

- In Extended Data Figure 1, I think the letters printed over the squares in panels D and E are incorrect; I think the letters in panel D should be F, G, and H, and the letters in panel E should be I and J. The figure legend should explain what these letters indicate.

- It would be helpful to see how the enrichment values correlate between replicate experiments. In Extended Data Figure 1, in addition to showing the correlation of read counts between replicates, the authors should show the correlation between the calculated degree of enrichment.

- In Extended Data Figure 4, the authors follow the behavior of “dosage-sensitive genes.” If they are using designations from a different paper, the authors should provide a citation in the figure legend. It would also be helpful to provide a brief explanation of the experiment that was used to define genes as dosage sensitive.

- Extended figure 5F says that “asterisks indicate significant differences compared to NaCl,” but I do not see any asterisks? There are also no error bars.

Suggestions

- The circle plots (eg. Fig. 1D) are not very informative. The chromosomal coordinates aren’t useful information. It is not possible to tell what the median value is, etc. It is not clear to me that the circularity improves visualization over a linear layout. A scatterplot of values seems like it would convey more information.

- I think the authors invent too many acronyms over the course of the manuscript (ADOPT, DoE, GOFA, ADOPT 2.0, ADOPT 2.1). This is especially confusing when the name of the acronym isn’t very informative. For instance, “autonomous dosage-optimization using plasmids with two-micron origin” is not at all a clear description of the method, and the word “ADOPT” has no relation to what it is describing. Please consider switching to standard informative terms like “overexpression profiling” for “ADOPT” and “hits” for “GOFAs.”

- There are various grammatical errors or word choice issues. A few points I noticed:

o “with more than 90%” should be “to more than 90%” (line 101).

o “Heat-shock responsible” should be “heat-shock responsive” (line 123)

o “they were excluded” should probably be something like “they dropped out” (line 127)

o The semicolon in line 153 should be a comma.

o The description of the mutations recovered in lines 164-5 is awkward; at the very least there should probably be an “and” before “G428.”

o “Pluses” in line 174 should be “pulses.”

o “Grey and orange lines show the empty vector as control and targets, respectively,” (line 175) would be easier to understand as “Grey lines show the empty vector as control and orange lines show the targets.”

o “Variance” in line 195 should be “variants.”

o The first “and” should be dropped from “… W303, and CEN.PK2-1C and a European wine strain, DBVP6765” (line 218)

o “Salt tolerant” should be hyphenated (line 274).

Reviewer #2: The work by Saeki et al, aims to identify overexpressed genes that can buffer or rescue the phenotype in yeast cells. For this, they generated a nice approach where a plasmid library is constructed to overexpress all the genes in the yeast genome. Based on this assay, they found different genes that improve the lab strain phenotype under stress conditions. The authors, particularly focus on salt stress tolerance, where they found genes that respond to the stress without the addition of calcium and/or potassium. Many of these genes were mitochondrial genes, providing new insights into the cellular response under high salt concentration. Overall, I think the conclusions are well supported by the data, and the authors performed a wealthy set of experiments to demonstrate their conclusions. Still, I have some major and minor comments that could help the authors to improve their manuscript and make it more understandable for a broader audience.

Major comments

1. I think the authors should include a better background overview of yeast traits and how these polygenic are influenced by different genes. There are several studies in the gene deletion collection and QTL mapping studies providing evidence of the genetics underlying these traits. That being said, what’s missing? What’s the motivation for your study. I believe this is missing in your INTRODUCTION section and is dispersed across the results section.

2. Throughout the results section different comparisons are made, but the statistics are generally missing. Either correlations, or comparisons between pools do not contain the statistics in the text or figures. Please include them every time to make sure that differences are significant (i.e L221, L227…).

3. Based on their G x E data, the authors could estimate for each overexpressed gene from their assay the estimated effect on each case using a linear model: lm (OE~environment x strain). In this way, for each gene they could actually assess G x E based on their title

L49. Protect what cell function?

L54. Please provide examples.

L62. Please expand on polygenic traits and how genes and genetic interactions underlie different phenotypes (such as heat or NaCl)

L117-120. Examples of genes are given, why those ones? Function of DoE?

L132. Remove ‘first’ if no other ‘second’ or ‘next’ is mentioned in this section.

L156. Authors mentioned that Ca+ increases the growth rate, but when observing figure2E I see something different, growth rate goes down. Am I missing something?

L164. Normally these mutations are expressed as ‘P393Q’, and so on.

L194-197. Please specify backgrounds and include this information in the introduction section

L238. Please indicate that a smaller number of generations was considered in this case.

L240. Authors should consider the greater overlap between CEN.PK and DBVPG6765

L243. Sometime is better to use the term ‘genetic background’ than ‘strain’.

L339. Authors should make sure that these part of the story is focused on mutants, and the lack of certain genes.

L365. Please describe how did you ‘observed’ mitochondria.

L404. Rephrase ‘only about 20%’. Was significantly reduced only 20%?

L526. Remove RNA-Seq

L530. Not sure these should be in methods.

Reviewer #3: In this manuscript, Saeki and colleagues investigate the contribution of gene overexpression in stressful environments for a better understanding of adaptive processes. They have developed an elegant strategy of overexpression profiling in yeast, named ADOPT. This method relies on libraries of Saccharomyces cerevisiae with each of the genes of the species cloned on a 2-micron plasmid, that are grown in competitive culture. Directed high-throughput long reads sequencing allows to estimate the degree of enrichment of the plasmid inserts in the final culture. This strategy allows for the detection of the so-called GOFAs (for genes whose overexpression is functionally adaptive), that the authors showed to be dependent of the backgrounds and environmental factors.

Data related to the most studied stresses in yeast (heat, salt and oxidative stresses) are presented and the salt stress was then dissected.

Overall the paper is clear and well written. The main strategy and its derivatives are well described, and the dissection of the salt tolerance lead to interesting results related to how strains can compensate through gene overexpression.

However, regarding the form of the manuscript, I find the introduction really weak, and the authors should make an effort to present a more complete state-of-the-art on their topic. Both aspects of adaptive processes (through gene overexpression, but not only) in yeast and strategies developed to test (genome-wide) gene over-expression should at least be developed.

Regarding the data and results per se, I have a major concern, as well as few minor comments that are presented below.

Major comments:

As mentioned by the authors, in some cases, control strains evolved and became more adapted than the “ADOPTed” ones (fig2G). This also suggests that, within the ADOPT libraries, primary mutations affecting a gene unrelated to the overexpressed one could lead to plasmid enrichment in the final culture. This factor is only poorly controlled in their experiments. The availability of replicates should be leveraged to counterpart this type of bias.

- there is no mention of the potential false positive GOFAs that might be highlighted, and why they would appear. This should be discussed at some point.

- l.107, the authors stated that “Enriched genes were highly reproducible across replicates”, while the plots they refers to do mostly not show that trend, in particular between the pools of different origins. This is even more visible in fig 2A.

- for most of the analyses, the authors consider as DoE the mean value of the 4 replicates. With this method, the reproducibility is not taken into account per se, and some bias related to the aforementioned phenomena may persist. The author should propose a way to seriously take the replicates into account in their strategy, or at least justify their choice.

Minor comments:

- l.51: "accidental increases": accidental is maybe not an appropriate word here. Please reformulate.

- l.129: why “unique”?

- l.133: Overexpression of NCS2 and NCS6 (-oe) independently resulted in increased heat resistance (Extended Data Fig. 2E and 2F): results are not significant for NCS2. This should be reformulated.

- l.143 and following: it is not clear here whether all the enriched genes were tested for growth rate estimation, or specifically the 4 genes mentioned. If all were not tested, please explain better the selection of the sets of genes, in particular YBR196C-A. And what are the hypotheses regarding the other genes? Are they considered as false positives? Why? It would be nice to have a short discussion about that.

- l.194 and following: the S288C strain has a cluster of 3 ENA genes, which may explain that, ENA1 was not detected as a GOFA in this background. This could be clearly stated in the manuscript. Regarding the other selected strains, for which genome sequencing data may be publicly available, it would be nice to make a parallel between the number of copies of ENA in the genome and the detection of this gene as a GOFA.

- fig 3 E-G: why was not the same number of generations considered for all backgrounds?

- l.241: “Particularly, the ENA1 and HAL genes were identified as GOFAs in CEN.PK (and

DBVPG6765)” this sentence can lead to confusion. ENA1 was only detected in CEN.PK and it HLA5 is the only HAL genes that was detected. Please reformulate.

Few random typos noticed while reading the manuscript:

- Extended figure1: mislabeling of the regions for which scatterplots were shown (from D to H while it should be from F to J)

- l.120: repetition of (Fig. 1G and 1H)

- fig 2D. "The cytoplasmic Ca2+ pluses" instead of "pulses"

**Have all data underlying the figures and results presented in the manuscript been provided?**

Reviewer #1: **No: **According to the Methods section, the authors deposited DNA sequencing data and RNA-seq data in the DNA Data Bank of Japan. But it does not appear that they deposited the Oxford Nanopore sequencing data that underlies the bulk of the experimental results. It also does not appear that numerical data underlying experiments such as growth rate measurements, etc. are included as Supplementary Information.

Reviewer #2: Yes

Reviewer #3: Yes

PLOS authors have the option to publish the peer review history of their article (what does this mean?). If published, this will include your full peer review and any attached files.

Reviewer #1: No

Reviewer #2: No

Reviewer #3: No

---

## [Decision Letter · Decision Letter 1]

13 Mar 2023

Dear Dr Moriya,

Thank you very much for submitting your Research Article entitled 'Overexpression profiling reveals cellular requirements in the context of genetic backgrounds and environments' to PLOS Genetics.

The manuscript was fully evaluated at the editorial level and by independent peer reviewers. The manuscript is much improved after the first round of review. Nevertheless, one of the reviewers still asks for some minor changes which you should address.

We therefore ask you to modify the manuscript according to the review recommendations. Your revisions should address the specific points made by each reviewer.

Yours sincerely,

Joseph Schacherer, Ph.D.

Guest Editor

PLOS Genetics

Geraldine Butler

Section Editor

PLOS Genetics

Reviewer's Responses to Questions

**Comments to the Authors:**

Reviewer #1: The manuscript is much improved, and I appreciate your changes in response to the previous round of feedback. Some comments on the latest version:

1) I agree that performing a statistical test for whether or not a given strain changed in abundance over the course of a multi-generational co-competition is bound to return positive hits for essentially every strain when comparing to the original abundances. Still, I appreciate that you applied an FDR threshold. Rather, the statistical analysis that is more important to perform is whether the hits from one replicate of an experiment were obtained more often than expected by chance in the other replicates. Off the top of my head, I am not certain how this is performed with more than two replicate experiments, but with two replicates people often use hypergeometric tests. From your Venn diagrams it certainly seems that the degree of overlap is far more than expected by chance, and so if you do the statistical tests I think they are highly likely to return positive results. But, for instance, the statement, “Seventeen genes in CEN.PK and 13 genes in DBVPG6765 were likely to be GOFAs under salt stress, as they were identified as hits in all the triplicate experiments,” is not correct as stated; the 17 genes are likely to be GOFAs because they were identified in all the triplicate experiments AND there were very few hits identified in just one or two replicates, that is, there is much more overlap in hits than expected by chance.

2) The revised introduction includes more background about overexpression profiling, which is welcome. You cited three examples of recent overexpression profiling screens (citations 20-22); to this list I think you should add Arita et al. Mol Syst Biol 2021, Magtanog et al. Nat Biotech 2011, and Payen et al. PLoS Genet 2016 (and perhaps others). In particular, Payen studied overexpression mutants that increased in abundance, and thus statements like, “However, those libraries have never been used to isolate beneficial genes” are incorrect. It would also be appropriate to mention that Robinson et al. tested the effects of overexpression in multiple strain backgrounds (lines 96-97).

3) I like the linear scatterplot visualization to showing the hits from the overexpression profiling. However, the x-axis is not labelled, and I cannot tell what the x-axis should be, such as in figure 1D-H.

4) If I understand correctly, pool_a and pool_b are replicate pools generated from the same arrayed strains generated in Makanae et al.? If that’s right, I would not refer to them as “independent” as on line 137, as any mutation, etc. present in the Makanae arrayed strains will be present in both pool_ a and pool_b. You say in line 155-157 that “if the same overexpression strain is enriched after multiple independent cultures, it strongly suggests that the overexpression strain was adaptive rather than resulting from mutation.” This would be true if the pools were truly independent, but as mentioned above, if the pools involve strains that had a common progenitor, any mutations experienced by those progenitors will be shared in the pools. I do not think you need to redo the experiments or analyses due to this concern, but it would be best to acknowledge it, either in the Results or the Discussion. The fact that you obtain hits from related but independent overexpression strains (such as NCS6 and NCS2) gives confidence that the results are not predominantly arising from shared secondary mutations.

There are still a lot of language issues that will need to be addressed. Here are some that I caught:

1) Line 96: “Previous” should have a lowercase “p.”

2) Line 137: You say the overexpression pools were maintained as a “liquid stock in a 96-well format”: do you mean as a frozen stock?

3) Line 178: When you say “as expected, the dosage sensitive genes identified in our previous studies […] were omitted,” I think “omitted” may be being misused in this context. I would interpret “omitted” to mean you removed them from consideration. Do you mean that those genes declined in abundance?

4) Line 201: Probably you should specify that the growth rate was increased by 1.29-fold in 40C, if that is what you meant.

5) Line 255: Remove the comma after “ENA1.”

6) Line 320: I think you’re missing a word after “were no longer.”

7) Line 322: I think you want to remove the word “under.”

8) Line 333: Hyphenate “salt-tolerant.”

9) Line 466: Overexpression profiling is not a new experimental system.

10) Line 585: I think you’re missing a word after “pooled libraries were.”

11) Line 590: Capitalize “the.”

12) Line 594: I don’t understand the sentence that begins “The mixed PCR products of and introduced into […]”.

13) Line 599: I think you mean “plates” instead of “templates.”

Reviewer #3: The authors have addressed all my comments. I do not have any additional suggestion.

**Have all data underlying the figures and results presented in the manuscript been provided?**

Reviewer #1: **No: **The authors say that large-scale datasets are in the process of being deposited, which is good, but it would appear they have not yet been deposited.

Reviewer #3: Yes

PLOS authors have the option to publish the peer review history of their article (what does this mean?). If published, this will include your full peer review and any attached files.

Reviewer #1: No

Reviewer #3: No

---

## [Editor Report · Decision Letter 2]

4 Apr 2023

Dear Dear Dr Moriya,

We are pleased to inform you that your manuscript entitled "Overexpression profiling reveals cellular requirements in the context of genetic backgrounds and environments" has been editorially accepted for publication in PLOS Genetics. Congratulations!

Yours sincerely,

Joseph Schacherer, Ph.D.

Guest Editor

PLOS Genetics

Geraldine Butler

Section Editor

PLOS Genetics

Comments from the reviewers (if applicable):

**Data Deposition**

http://datadryad.org/submit?journalID=pgenetics&manu=PGENETICS-D-22-01127R2

**Press Queries**

---

## [Editor Report · Acceptance letter]

24 Apr 2023

PGENETICS-D-22-01127R2 

Overexpression profiling reveals cellular requirements in the context of genetic backgrounds and environments 

Dear Dr Moriya, 

We are pleased to inform you that your manuscript entitled "Overexpression profiling reveals cellular requirements in the context of genetic backgrounds and environments" has been formally accepted for publication in PLOS Genetics! Your manuscript is now with our production department and you will be notified of the publication date in due course.

With kind regards,

Zsofia Freund

PLOS Genetics

On behalf of:
